# Weighted ROC Curve in Cost Space:
# Extending AUC to Cost-Sensitive Learning

**Huiyang Shao**[1,2]       **Qianqian Xu**[1*]       **Zhiyong Yang**[2]
**Peisong Wen**[1,2]       **Peifeng Gao**[2]       **Qingming Huang**[2,1,3*]
[1] Key Lab. of Intelligent Information Processing, Institute of Computing Tech., CAS
[2] School of Computer Science and Tech., University of Chinese Academy of Sciences
[3] BDKM, University of Chinese Academy of Sciences
shaohuiyang21@mails.ucas.ac.cn xuqianqian@ict.ac.cn
wenpeisong20z@ict.ac.cn gaopeifeng21@mails.ucas.ac.cn
yangzhiyong21@ucas.ac.cn qmhuang@ucas.ac.cn

## Abstract

In this paper, we aim to tackle flexible cost requirements for long-tail datasets, where we need to construct a (1) cost-sensitive and (2) class-distribution robust learning framework. The misclassification cost and the area under the ROC curve (AUC) are popular metrics for (1) and (2), respectively. However, limited by their formulations, models trained with AUC are not well-suited for cost-sensitive decision problems, and models trained with fixed costs are sensitive to the class distribution shift. To address this issue, we present a new setting where costs are treated like a dataset to deal with arbitrarily unknown cost distributions. Moreover, we propose a novel weighted version of AUC where the cost distribution can be integrated into its calculation through decision thresholds. To formulate this setting, we propose a novel bilevel paradigm to bridge weighted AUC (WAUC) and cost. The inner-level problem approximates the optimal threshold from sampling costs, and the outer-level problem minimizes the WAUC loss over the optimal threshold distribution. To optimize this bilevel paradigm, we employ a stochastic optimization algorithm (SACCL) which enjoys the same convergence rate ($O(\epsilon^{-4})$) with the SGD. Finally, experiment results show that our algorithm performs better than existing cost-sensitive learning methods and two-stage AUC decisions approach.

## 1 Introduction

Receiver Operating Characteristics (ROC) is a popular tool to describe the trade-off between the True Positive Rate (TPR) and False Positive Rate (FPR) of a scoring function. AUC is defined by the area under the ROC curve [17, 18]. This metric naturally measures the average classification performance under different thresholds and is widely used (*e.g.*, disease prediction [19], and anomaly detection [29]). Compared with accuracy, AUC is insensitive to the threshold and cost [7], making it be a popular metric for long-tail learning [32] and achieve remarkable success [24, 44, 26].

Similar to AUC optimization, cost-sensitive learning is a common data mining method [10, 2, 4]. The main goal is to incorporate the misclassification costs in the model, which is more compatible with realistic scenarios (*e.g.*, the cost of misdiagnosing a disease as healthy is greater than the counterexample). Over the past two decades, researchers have pointed out that the ROC curve can be transferred to cost space by utilizing a threshold choice method, this is equivalent to computing the area under the convex hull of the ROC curve [21]. In this way, AUC can be seen as the performance of the model with a uniform cost distribution [16]. However, AUC considers all situations, which can not focus more on hard samples, Partial AUC (PAUC) is proposed as an extension of AUC with

---

[*]Corresponding authors.

37th Conference on Neural Information Processing Systems (NeurIPS 2023).

Table 1: Comparison with existing classification settings. Cost distribution represents the cost condition of each setting.

| Different setting | Formulation | Attr.1 | Attr.2 | Cost distribution |
|---|---|---|---|---|
| Cost learning | $\mathbb{E}_x[c \cdot \pi \cdot p(1\|x) + (1-c) \cdot (1-\pi) \cdot p(0\|x)]$ | ✗ | ✗ | |
| AUC/PAUC | $\mathbb{E}_\tau[\text{TPR}(\tau)\text{FPR}'(\tau)] \quad \tau \sim U(a,b)$ | ✓ | ✗ | |
| WAUC | $\mathbb{E}_\tau[\text{TPR}(\tau)\text{FPR}'(\tau)W(\tau)] \quad W(\tau) \sim beta(a,b)$ | ✓ | ✗ | |
| Our method | $\mathbb{E}_{\tau \sim \boldsymbol{\tau}^*}[\text{TPR}(\tau)\text{FPR}'(\tau)] \quad \boldsymbol{\tau}^* \in \arg\min_\tau \mathcal{L}_{COST}$ | ✓ | ✓ | |

truncated uniform cost distribution [31]. Recently, some studies extend PAUC using parameterized cost distributions and propose WAUC to fit real-world applications [16, 30].

Whether we use AUC or cost learning, our main purpose is to train models with these attributes:

**Attr.1**: The trained model can be robust to class distribution shift in the test **without class prior**.

**Attr.2**: The trained model can be robust to cost distribution in the test **without cost prior**.

However, to the best of our knowledge, there are few methods can train a model to have both of these attributes. According to Tab. 1, both AUC-related methods and cost-sensitive learning require a strong prior knowledge of the cost distribution; **(1)** The cost learning mainly considers a specified cost $c$ and class imbalanced radio $\pi$. Models trained under this method are sensitive to class distribution, which does not apply to the scenario where test data distribution with offset. **(2)** AUC (PAUC) assumes that the cost distribution belongs to (truncated) uniform cost distribution $U(a,b)$. Models trained with them will have poor performance when the true cost distribution is not uniform [16]. **(3)** WAUC considers optimizing models based on more complex forms of cost distribution, such as $beta(a,b)$. However, we can not obtain the cost prior in real problem scenarios, *e.g.*, financial market prediction [14]. Considering the weakness comes from the existing settings, we will explore the following question in this paper:

> *Can we bridge AUC and complicated cost distribution to training robust model on desired cost-sensitive and arbitrary class imbalanced decision scenarios?*

To answer this question, **we propose a view that, in some real applications [14], the cost, like the instance data, is not available prior but can be obtained by sampling.** Therefore, we choose to sample desired cost to approximate the true cost distribution. Different from previous settings, ours is closer to real world, the main process can be divided into three parts:

**Step.1 Cost Sampling:** Firstly, we sample some desired costs to construct the empirical cost set.

**Step.2 Data Sampling:** Next, we sample some instance data to construct the empirical dataset.

**Step.3 Build Formulation:** Finally, we construct the appropriate formulation to maximize the performance in different desired costs and ensure model is robust to distribution shift.

It is natural for us to ask the question: Can we use the existing methods to realize this process? It's clear the answer is no. For AUC-related methods, they can not perform Step.1, and for cost-sensitive learning, they fail to achieve robust distribution shift and multiple costs in Step.3 (as shown in Fig. 1 (orange line). Hence, we propose a novel bilevel formulation combining the advantages of WAUC and cost learning. The inner-level process calculates the optimal threshold from sampling costs, and the outer-level process minimizes the WAUC loss over the optimal threshold distribution. The method can help the model improve robustness to class distribution in cost-sensitive decision problems. The main process is shown in Fig. 1 (green line). We summarize our contributions below:

- We propose a setting that focuses on the robustness of the model to the class distribution and cost distribution simultaneously. This setting treats cost as data that can be sampled, not as prior information, which is closer to the real-world cost-sensitive scenario.

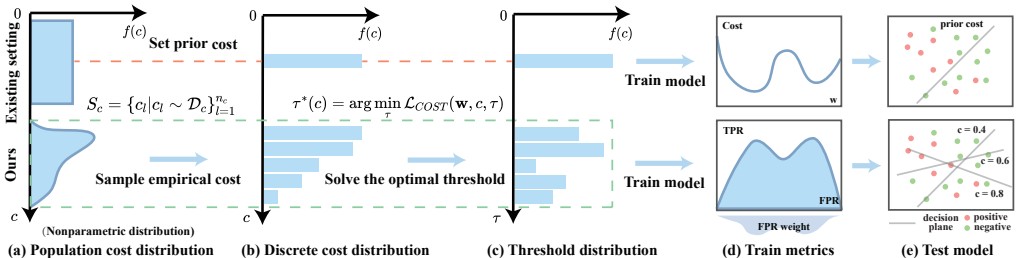

Figure 1: The comparison of our proposed setting with the previous setting. The orange line represents the previous cost-sensitive learning approach, and the green line represents our method.

- We present a bilevel paradigm where the inner cost function is an inner constraint of outer WAUC optimization. For sake of optimization, we reformulate this paradigm into a nonconvex-strongly convex bilevel form. Moreover, we employ a stochastic optimization algorithm for WAUC (SACCL), which can solve this problem efficiently.
- We conduct extensive experiments on multiple imbalanced cost-sensitive classification tasks. The experimental results speak to the effectiveness of our proposed methods.

## 2 Observation and Motivation

In Tab. 1, we compare the existing methods with ours from different views. However, the table comparison does not have a very visual presentation. In this section, we will analyze the disadvantages of existing settings and explain our motivation. We train the model with all methods on a Cifar-10-Long-Tail training set under the different imbalanced ratios and cost distribution. We visualize the feature representation (the last layer's output of the trained model) in test data by t-SNE. The blue point represents negative samples predicted by the optimal threshold, and the orange point represents positive samples. The smaller the overlap between them, the better the performance. From Fig. 2, we can make the following remarks: **(1)** According to Fig. 2 (a), AUC is robust to changes in the imbalance ratio but completely not applicable with the cost distribution. **(2)** According to Fig. 2 (b), cost learning can process different cost distributions, but is sensitive to imbalance ratios.

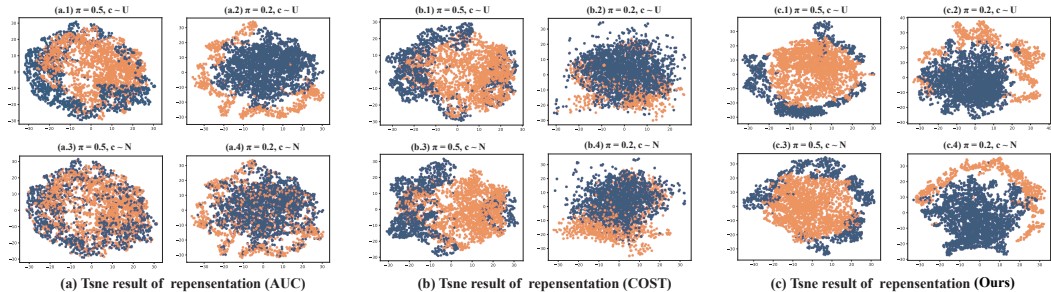

Figure 2: The feature representations comparison among different methods. (a) AUC optimization. (b) Cost learning. (c) Ours result. We solve the optimal threshold with the $\hat{\mathcal{L}}_{COST}$ (defined in Sec. 3). $\pi$ denotes the probability of positive class, $c$ denotes the cost ratio of misclassification ($U$ denotes Uniform, $N$ denotes Normal). For example, $\pi = 0.5$, $c \sim N$ means model tested on dataset which has imbalanced ratio $\pi$ and cost set sampled from $N$.

Hence, our motivation is to propose a new approach to solve the problems in AUC and Cost-sensitive learning. As shown in Fig. 2 (c), our proposed method can have better learning results in various complex cost distributions and imbalance ratios. That means our method can overcome the shortcomings of traditional AUC and cost-learning, which perfectly fits the proposed setting.

## 3 Preliminaries

**Notations.** In this section, we give definitions and preliminaries about AUC. We denote $(\boldsymbol{x}, y)$ be an instance, where $\boldsymbol{x}$ is drawn from feature space $\mathcal{X} \subseteq \mathbb{R}^d$ ($d$ is feature number) and $y$ is drawn from

label space $\mathcal{Y} = \{0, 1\}$. Let $\mathcal{D}_\mathcal{P}$ ($\mathcal{D}_\mathcal{N}$ *resp.*) be positive (negative *resp.*) instance distribution. Let $\boldsymbol{x}^+ \sim \mathcal{D}_\mathcal{P}$ ($\boldsymbol{x}^- \sim \mathcal{D}_\mathcal{N}$ *resp.*) be positive (negative *resp.*) instance. We denote $S_+ = \{(\boldsymbol{x}_i^+, y_i)\}_{i=1}^{n_+}$ ($S_- = \{(\boldsymbol{x}_j^-, y_j)\}_{j=1}^{n_-}$ *resp.*) as a set of training data drawn from $\mathcal{D}_\mathcal{P}$ ($\mathcal{D}_\mathcal{N}$ *resp.*), where $n_+$ ($n_-$ *resp.*) denotes the instance number of $S_+$ ($S_-$ *resp.*). Let $\mathbb{I}_{(\cdot)}$ be the indicator function, which returns 1 when the condition is true and 0 otherwise. In this paper, we focus on the deep neural network scoring function $s(\boldsymbol{w}, \boldsymbol{x}) : \mathcal{X} \mapsto [0, 1]$, where parameterized by $\boldsymbol{w}$ on an input $\boldsymbol{x}$.

**AUC & WAUC.** For specified threshold $\tau$, the TPR of a classifier derived from $s(\boldsymbol{w}, \boldsymbol{x})$ measures the likelihood that it accurately predicts a positive instance when getting a random positive instance from $\mathcal{D}_\mathcal{P}$. Formally, we have:

$$\text{(Pop.) } \text{TPR}_s(\tau) = \mathbb{P}_{\boldsymbol{x}^+ \sim \mathcal{D}_\mathcal{P}}[s(\boldsymbol{w}, \boldsymbol{x}^+) > \tau] \quad \text{(Emp.) } \widehat{\text{TPR}}_s(\tau) = \frac{1}{n_+} \sum_{i=1}^{n_+} \mathbb{I}_{s(\boldsymbol{w}, \boldsymbol{x}_i^+) > \tau}. \quad (1)$$

In a similar spirit, the classifier's FPR on threshold $\tau$ refers to the probability that it predicts positive when it gets a negative instance from $\mathcal{D}_\mathcal{N}$.

$$\text{(Pop.) } \text{FPR}_s(\tau) = \mathbb{P}_{\boldsymbol{x}^- \sim \mathcal{D}_\mathcal{N}}[s(\boldsymbol{w}, \boldsymbol{x}^-) > \tau] \quad \text{(Emp.) } \widehat{\text{FPR}}_s(\tau) = \frac{1}{n_-} \sum_{j=1}^{n_-} \mathbb{I}_{s(\boldsymbol{w}, \boldsymbol{x}_j^-) > \tau}. \quad (2)$$

AUC measures a scoring function's trade-off between TPR and FPR under uniform thresholds. Denote $\tau$ drawn from the distribution $\mathcal{D}_\tau$, WAUC utilizes the threshold distribution explicitly based on AUC. $\text{FPR}'_s$ denotes the probability density function of $s(\boldsymbol{w}, \boldsymbol{x}^-)$.

$$\text{AUC} = \int_\infty^{-\infty} \text{TPR}_s(\tau) \text{FPR}'_s(\tau) d\tau \quad (3a)$$

$$\text{WAUC} = \int_\infty^{-\infty} \text{TPR}_s(\tau) \text{FPR}'_s(\tau) p(\tau) d\tau, \quad (3b)$$

**Cost function [2].** In some real application scenarios, we need to consider the misclassification cost. We denote $c_{(\cdot)}$ as misclassification cost for class $(\cdot)$, cost $c$ drawn from $\mathcal{D}_c$. Since we could not obtain the cost distribution $\mathcal{D}_c$, we sample empirical set $S_c = \{c_l\}_{l=1}^{n_c}$, $n_c$ denotes the sample number of cost $c$. Given a scoring function $s$ and parameter $\boldsymbol{w}$, the cost function $\mathcal{L}_{COST}$ is (the empirical version of cost function, $\widehat{\mathcal{L}}_{COST}$ contains the empirical forms of TPR and FPR):

$$\mathcal{L}_{COST}(\boldsymbol{w}, c, \tau^*(c)) = c \cdot \pi \cdot (1 - \text{TPR}_s(\tau^*(c))) + (1 - c) \cdot (1 - \pi) \cdot \text{FPR}_s(\tau^*(c)), \quad (4)$$

where $\pi = n_+/(n_+ + n_-)$, $c = c_+/(c_+ + c_-)$ and $\tau^*(c)$ is optimal threshold for score function $s$ under specified $c$ [21], the sample number $n_\tau = n_c$.

## 4 Problem Formulation

In this section, we introduce how to link the ROC curve to the cost space. First, we reformulate Eq.(3) into expectation:

$$\text{AUC} = \underset{\tau \sim U}{\mathbb{E}} \left[ \text{TPR}_s(\tau) \cdot \text{FPR}'_s(\tau) \right] \quad (5a)$$

$$\text{WAUC} = \underset{\tau \sim \mathcal{D}_\tau}{\mathbb{E}} \left[ \text{TPR}_s(\tau) \cdot \text{FPR}'_s(\tau) \right]. \quad (5b)$$

If threshold $\tau$ is drawn from the uniform distribution $U$, WAUC will degrade to the standard AUC formulation. However, AUC only describes the global mean performance under all possible costs. If we want to extend AUC to cost-sensitive problems, maybe lifting the restriction on the uniform distribution of $\tau$ is a good solution. Hence, we release the $\mathcal{D}_\tau$'s restriction to make it belongs to complicated distribution (*e.g.*, normal distribution, exponential distribution). Then we can extend AUC to WAUC. However, using WAUC raises another question: how do we get $\mathcal{D}_\tau$? We find that $\tau^*(c)$ is one of parameters of $\mathcal{L}_{COST}(\boldsymbol{w}, c, \tau^*(c))$. A natural idea is to use $\mathcal{L}_{COST}$ to solve for the optimal $\tau^*(c)$ and to combine the $\tau^*(c)$ solved for at different $c$ to obtain $\mathcal{D}_\tau$.

$$\tau^*(c) = \arg\min_\tau \mathcal{L}_{COST}(\boldsymbol{w}, \tau, c) = c \cdot \pi \cdot (1 - \text{TPR}_s(\tau)) + (1 - c) \cdot (1 - \pi) \cdot \text{FPR}_s(\tau), \quad (6)$$

If we couple Eq.(5b) and Eq.(6) together so that WAUC can enjoy the optimal threshold distribution in $\mathcal{L}_{COST}$, then we can break the barrier between the ROC curve and the cost space. With the help of the threshold as a bridge, we can extend the AUC metric to achieve the WAUC cost-sensitive learning. Then we give the problem formulation (intuitively, from the result of 2 (c), (OP0) satisfies both Attr.1 and Attr.2 simultaneously):

$$(OP0) \quad \text{(outer.)} \qquad \text{WAUC} = \mathop{\mathbb{E}}_{\tau \sim \boldsymbol{\tau}^*} \left[ \text{TPR}_s(\tau) \cdot \text{FPR}_s'(\tau) \right]$$
$$\text{(inner.)} \quad \boldsymbol{\tau}^* = \{\tau^*(c) = \arg\min_\tau \mathcal{L}_{COST}(\boldsymbol{w}, \tau, c) | c \sim \mathcal{D}_c \} \tag{7}$$

Nevertheless, there are still three main challenges in WAUC cost-sensitive learning:

**(1)** Given the scoring function $s$ and negative dataset $S_-$, how to estimate $\text{FPR}_s'(\tau)$ in WAUC?

**(2)** The inner problem is nonconvex, which is hard to give a theoretical convergence guarantee.

**(3)** How to design a formulation that can bridge WAUC and $\mathcal{L}_{COST}$ so that WAUC can be optimized over the cost distribution of the desired problem scenario?

We will address the challenge **(1)** in Sec. 5.1, challenge **(2)** in Sec. 5.2 and challenge **(3)** in Sec. 5.3.

## 5 Methodology

### 5.1 The Estimation of False Positive Rate

For challenge **(1)**, we choose the kernel density estimation (KDE) to estimate $\text{FPR}_s'(\tau)$ and denote it as $K(x)$ (please see definition in Sec. C.1). Then we can address the density estimation problem. However, Eq.(5b) still exists non-differentiable and non-smooth term $\mathbb{I}_{(\cdot)}$, which is hard to optimize. Hence, we propose the following smooth and differentiable WAUC estimator to approximate Eq.(5b).

**Definition 5.1.** Denote $K(x)$ be statistics kernel with bandwidth $m$ and $S_-^{\boldsymbol{w}} = \{s(\boldsymbol{w}, \boldsymbol{x}_j^-)\}_{j=1}^{n_-}$. With Lemma 5.2, we have the approximate estimator and loss function for WAUC:

$$\widehat{\text{WAUC}} = \int_\infty^{-\infty} \text{TPR}_s(\tau) \mathcal{K}(S_-^{\boldsymbol{w}}, \tau) p(\tau) d\tau, \quad \widehat{\mathcal{L}}_{\text{WAUC}}(\boldsymbol{w}, \boldsymbol{\tau}) = \frac{1}{n_\tau} \sum_{l=1}^{n_\tau} \hat{h}(\boldsymbol{w}, \tau_l) \tag{8}$$

where $\boldsymbol{\tau} = \{\tau_l\}_{l=1}^{n_\tau}$ and the point loss $\hat{h}$ is defined by

$$\hat{h}(\boldsymbol{w}, \tau) = 1 - \frac{1}{n_+ n_-} \sum_{i=1}^{n_+} \sum_{j=1}^{n_-} \sigma(s(\boldsymbol{w}, \boldsymbol{x}_i^+) - \tau_l) \cdot K((s(\boldsymbol{w}, \boldsymbol{x}_j^-) - \tau_l)/m)/m. \tag{9}$$

$\sigma(x) = 1/(1 + \exp(-\beta x))$, $\beta$ is smooth parameter and we have $\sigma(x) \xrightarrow{\beta \to \infty} \mathbb{I}_x$.

**Lemma 5.2.** *Given a scoring function $s$, if $\boldsymbol{\tau}$ is known, when the number of instances is large enough, $\widehat{\text{WAUC}}$ almost surely converges to WAUC.*

$$\lim_{n_- \to \infty} |\widehat{\text{WAUC}} - \text{WAUC}| \xrightarrow{a.s.} 0. \tag{10}$$

With KDE consistency [41], when the negative sample size is large enough, Lemma 5.2 provides theoretical approximation guarantees for our proposed WAUC estimator in Prop. 5.1.

### 5.2 The Estimation of Threshold Weighting

For challenge **(2)**, a natural idea is to use $\widehat{\mathcal{L}}_{COST}$ to solve for the optimal threshold set $\hat{\boldsymbol{\tau}}^*$ when given the cost set $S_c$ and the scoring function $s$. Then we can use the optimal threshold set $\hat{\boldsymbol{\tau}}^*$ to calculate $\widehat{\text{WAUC}}$. Firstly, we define the solution for $\hat{\boldsymbol{\tau}}^*$ be

$$\hat{\boldsymbol{\tau}}^* = \left\{ \hat{\tau}^*(c) | \hat{\tau}^*(c) \in \arg\min_\tau \widehat{\mathcal{L}}_{COST}(\boldsymbol{w}, c), c \in S_c \right\}. \tag{11}$$

However, it's noticed that the $\arg\min_\tau \widehat{\mathcal{L}}_{COST}$ in Eq.(11) is non-convex. As we analyzed before, $\boldsymbol{\tau}^*$ is the inner constraint of $\widehat{\mathrm{WAUC}}$. To the best of our knowledge, there are few studies on optimizing two coupled non-convex problems simultaneously with theoretical convergence guarantees. Most studies on coupled optimization assume that the inner problems have good properties, such as strong convexity. Hence, we propose the approximated convex formulation of the inner problem for $\hat{\boldsymbol{\tau}}^*$.

**Theorem 5.3.** *When we set $\kappa, M$ are large positive numbers and $M'^2 < M^2 \frac{6\kappa^2 e^{3\kappa}}{(e^\kappa+1)^6}$, then we have the approximated convex formulation for $\hat{\mathcal{L}}_{COST}$*

$$\min_{\tau, \boldsymbol{P} \in \mathbb{R}^{n_+}, \boldsymbol{N} \in \mathbb{R}^{n_-}} \widehat{\mathcal{L}}_{eq}(\boldsymbol{w}, \tau, c) := c \cdot \pi \cdot (1 - \frac{1}{n_+}\sum_{i=1}^{n_+} P_i) + (1-c)\cdot(1-\pi)\cdot(\frac{1}{n_-}\sum_{j=1}^{n_-} N_j)$$

$$+ \frac{1}{n_+}\sum_{i=1}^{n_+} M'\psi(s(\boldsymbol{w}, \boldsymbol{x}_i^+) - \tau) - P_i(s(\boldsymbol{w}, \boldsymbol{x}_i^+) - \tau)) + M\psi(P_i - 1) + M\psi(\tau - 1) \quad (12)$$

$$+ \frac{1}{n_-}\sum_{j=1}^{n_-} M'\psi(s(\boldsymbol{w}, \boldsymbol{x}_j^-) - \tau) - N_j(s(\boldsymbol{w}, \boldsymbol{x}_j^-) - \tau)) + M\psi(N_j - 1) \ \ 0 \le \tau, P_i, N_j$$

*where $\psi(x) = \log(1 + \exp(\kappa x))/\kappa$. $\widehat{\mathcal{L}}_{eq}$ in Eq.(12) is $\mu_g$-strongly convex w.r.t. $\boldsymbol{\tau}$. Eq.(12) has same solution as $\min_\tau \widehat{\mathcal{L}}_{COST}$ when the parameters satisfy the conditions of the penalty.*

Thm. 5.3 provides an optimization method with good properties. Eq. (12) adopts the penalty function to convert inequality constraints into a part of the objective function. When these inequality constraints are not satisfied, the objective function will increase to infinity. Otherwise, we will get $P_i = \mathbb{I}[s(\boldsymbol{w}, x_i^+) > \tau]$ and $N_j = \mathbb{I}[s(\boldsymbol{w}, x_j^-) > \tau]$, then we will get the same formulation as $\widehat{\mathcal{L}}_{COST}$. When the parameters meet the requirements, Eq.(12) has the same solution as $\widehat{\mathcal{L}}_{COST}$. We give the proof of Thm. 5.3 and the definition $\mu$ in Sec. C.4. Moreover, we give the analysis of approximation error between $\widehat{\mathcal{L}}_{COST}$ and Thm. 5.3 in Sec. B.7.

## 5.3 Bilevel Optimization for WAUC learning

After answering questions in challenge **(1)** and **(2)**, we have solved most of the problems in WAUC cost-sensitive learning. However, there remains a challenge **(3)** in optimization: How do we design learning paradigms to solve the coupled optimization problem of WAUC and $\mathcal{L}_{COST}$? In recent years, bilevel optimization has achieved remarkable success. This approach can combine two related optimization problems to form a coupled optimization formulation. Hence, with Prop. 5.1 and Thm.5.3, we propose a bilevel paradigm to formulate this coupled optimization problem.

$$(OP1) \ (\text{outer.}) \quad \min_{\boldsymbol{w}} \hat{F}(\boldsymbol{w}) := \hat{f}(\boldsymbol{w}, \boldsymbol{\tau}^*) := \widehat{\mathcal{L}}_{\mathrm{WAUC}}(\boldsymbol{w}, \hat{\boldsymbol{\tau}}^*)$$

$$(\text{inner.}) \quad \hat{\boldsymbol{\tau}}^* = \arg\min_{\boldsymbol{\tau}, \boldsymbol{P}_a, \boldsymbol{N}_a} \hat{g}(\boldsymbol{w}, \boldsymbol{\tau}) := \frac{1}{n_\tau}\sum_{l=1}^{n_\tau} \widehat{\mathcal{L}}_{eq}(\boldsymbol{w}, \tau_l, c_l), \quad (13)$$

where $\boldsymbol{P}_a \in \mathbb{R}^{n_\tau \times n_+}$ and $\boldsymbol{N}_a \in \mathbb{R}^{n_\tau \times n_-}$. $(OP1)$ describes a bilevel optimization formulation for WAUC cost-sensitive learning, where the inner-level provides a threshold optimization process, and the outer-level minimizes the WAUC loss over the optimal threshold distribution. Moreover, this formulation is consistent with the mainstream bilevel optimization problem (outer-level is smooth and non-convex, inner-level is convex and smooth), which enjoys a faster convergence rate.

# 6 Optimization Algorithm

In this section, we focus on optimizing $(OP1)$ in an end-to-end manner. Hence, we propose a stochastic algorithm for WAUC cost-sensitive learning shown in Alg. 1, which is referred to SAACL.

## 6.1 Main Idea of SAACL

We provide some intuitive explanations of our algorithm. At each iteration $k$, SACCL alternates between the inner-level gradient update on $\boldsymbol{\tau}$ and the outer-level gradient update on $\boldsymbol{w}$. During

---
**Algorithm 1** Stochastic Algorithm for WAUC Cost-sensitive Learning
---
**Input:** training data $S_+$ and $S_-$, iteration numbers $K$ and $T$, batch size $B$.
**Initialize:** parameters $\boldsymbol{w}_0 \in \mathbb{R}^n, \boldsymbol{\tau}_0 \in \mathbb{R}^{n_\tau}$, stepsizes $\alpha_k, \beta_k$.
**for** $k = 0$ **to** $K$ **do**
    set $\boldsymbol{\tau}_{k,0} = \boldsymbol{\tau}_k$.
    **for** $t = 0$ **to** $T$ **do**
        drawn $\mathcal{B}_t = \{(\boldsymbol{x}_b, y_b)\}_{b=1}^B$ from $S_+$ and $S_-$ uniformly.
        $\forall c_l \in S_c, \tau_{k,t+1}^l = \tau_{k,t}^l - \beta_k \nabla_{\boldsymbol{\tau}} \hat{g}(\boldsymbol{w}_k, c_l; \mathcal{B}_t)$.
    **end for**
    set $\boldsymbol{\tau}_{k+1} = \boldsymbol{\tau}_{k,T}$
    drawn $\mathcal{B}_k = \{(\boldsymbol{x}_b, y_b)\}_{b=1}^B$ from $S_+$ and $S_-$ uniformly.
    $\boldsymbol{w}_{k+1} = \boldsymbol{w}_k - \alpha_k[\nabla_{\boldsymbol{w}} \hat{f}(\boldsymbol{w}_k, \boldsymbol{\tau}_{k+1}; \mathcal{B}_k) - \nabla_{\boldsymbol{w}\boldsymbol{\tau}}^2 \hat{g}(\boldsymbol{w}_k, \boldsymbol{\tau}_{k+1}; \mathcal{B}_k)\cdot$
        $\left[\frac{N}{L_{g,1}} \prod_{n=1}^{N'} \left(I - \frac{1}{L_{g,1}} \nabla_{\boldsymbol{\tau}\boldsymbol{\tau}}^2 \hat{g}(\boldsymbol{w}, \boldsymbol{\tau}_{k+1}; \mathcal{B}_k)^{-1}\right)\right] \nabla_{\boldsymbol{\tau}} \hat{f}(\boldsymbol{w}_k, \boldsymbol{\tau}_{k+1}; \mathcal{B}_k)$
**end for**
---

iteration $k$, we update $\boldsymbol{\tau}_{k,t}$ with standard SGD $T$ steps to ensure that $\boldsymbol{\tau}_{k+1}$ is as optimal as possible. After updating inner-level variables, we perform outer-level optimization with $\boldsymbol{\tau}_{k+1}$ as the parameter to update $\boldsymbol{w}_k$. Notice that $T$ will not take a large value to ensure the validity of the coupling update of $\boldsymbol{\tau}$ and $\boldsymbol{w}$. Let $\alpha_k$ and $\beta_k$ be stepsizes of $\boldsymbol{w}$ and $\boldsymbol{\tau}$ that have the same decrease rate as SGD. We denote $n$ be the number of elements in deep neural network parameters $\boldsymbol{w}$.

### 6.2  Convergence Analysis of SAACL

In this subsection, we present the convergence analysis for SAACL. We give some Lipschitz continuity assumptions that are common in bilevel optimization problems [11, 28].

**Assumption 6.1. (Lipschitz continuity)** Assume that $f$, $\nabla f$, and $\nabla g$ are respectively $L_{f,0}, L_{f,1}$, $L_{g,1}$-Lipschitz continuous.

**Assumption 6.2. (Bounded stochastic derivatives)** The variance of stochastic derivatives $\nabla f(\boldsymbol{w}, \boldsymbol{\tau}; \mathcal{B})$ and $\nabla g(\boldsymbol{w}, \boldsymbol{\tau}; \mathcal{B})$ are bounded by $\sigma_{f,1}^2, \sigma_{g,1}^2$, respectively.

Based on Assumption 6.1 and Assumption 6.2, following [5], Thm. 6.3 indicates that we can optimize ($OP1$) with the same convergence rate as the traditional SGD algorithm.

**Theorem 6.3.** *Suppose Assumption 6.1 and 6.2 hold. We define*

$$\bar{\alpha}_1 = \frac{1}{2L_F + 4L_f L_y + 2L_f L_{yx}/(L_y \eta)}, \quad \bar{\alpha}_2 = \frac{16T\mu L_{g,1}}{(\mu + L_{g,1})^2 (8L_f L_y + 2\eta L_{yx} \tilde{C}_f^2 \bar{\alpha}_1)}, \quad (14)$$

*where $\eta = L_F/L_y$, $L_F$, $L_f$, $L_y$ and $\tilde{C}_f^2$ come from Lem. 2 and Lem. 4 in [5]. We select the following stepsize as*

$$\alpha_k = \min\left\{\bar{\alpha}_1, \bar{\alpha}_2, \frac{1}{\sqrt{K}}\right\} \qquad \beta_k = \frac{8L_f L_y + 2\eta L_{yx} \tilde{C}_f^2 \bar{\alpha}_1}{4T\mu} \alpha_k \qquad (15)$$

*For any $T \geq 1$, the iteration sequence $\{\boldsymbol{w}_k\}_{k=1}^K$ and $\{\boldsymbol{\tau}_k\}_{k=1}^K$ generated by Algorithm 1 satisfy*

$$\frac{1}{K} \sum_{k=0}^{K-1} \mathbb{E}\left[\|\nabla F(\boldsymbol{w}_k)\|^2\right] \leq \gamma \left(\frac{3M\kappa e^\kappa/(e^\kappa + 1)^2 + L_{g,1}}{24M\kappa e^\kappa/(e^\kappa + 1)^2 L_{g,1}}\right)^2 \frac{1}{T\sqrt{K}} + O\left(\frac{1}{\sqrt{K}}\right). \qquad (16)$$

*where $\gamma = 2\alpha\sigma_{g,1}^2 \frac{L_f}{L_y} \left(1 + 5L_f L_y \bar{\alpha}_1 + \frac{\eta L_{yx} \tilde{C}_f^2}{4} \bar{\alpha}_1^2\right) (8L_f L_y + 2\eta L_{yx} \tilde{C}_f^2 \bar{\alpha}_1)^2$*

*Remark* 6.4. When $\kappa$ and $M$ are large enough positive integers, according to Eq. (16), Alg. 1 is still guaranteed to find a $\epsilon$-stationary point within $O(\epsilon^{-4})$ iterations ($\epsilon$ is error tolerance).

## 7  Experiments

In this section, we conduct a series of experiments for WAUC cost-sensitive learning on common long-tail benchmark datasets. Due to space limitations, please refer to Sec. B for the details of our experiments. The source code is available in supplemental materials.

## 7.1 Dataset Details

We use three datasets: **Binary CIFAR-10-Long-Tail Dataset** [23], **Binary CIFAR-100-Long-Tail Dataset** [23], and **Jane Street Market Prediction** [14]. Binary CIFAR-10-Long-Tail Dataset and Binary CIFAR-100-Long-Tail Dataset are common datasets in long-tail learning, and we construct their cost distributions. Jane Street Market Prediction is data from real cost-sensitive learning application scenarios. For all datasets, we divide them into the training set, validation set, and test set with a proportion 0.7:0.15:0.15. All image data is normalized to ensure a more stable training process.

Table 2: Performance comparisons on benchmark datasets with different metrics. The first and second best results are highlighted with **bold text** and underline, respectively.

| dataset | type | methods | Subset1 | | Subset2 | | Subset3 | | $\widehat{AUC}\uparrow$ | | |
|---|---|---|---|---|---|---|---|---|---|---|---|
| | | | $\widehat{WAUC}\uparrow$ | $\widehat{\mathcal{L}}_{COST}\downarrow$ | $\widehat{WAUC}\uparrow$ | $\widehat{\mathcal{L}}_{COST}\downarrow$ | $\widehat{WAUC}\uparrow$ | $\widehat{\mathcal{L}}_{COST}\downarrow$ | Subset1 | Subset2 | Subset3 |
| CIFAR-10-LT | Competitors | BCE | 0.525 | 0.027 | 0.533 | 0.015 | 0.318 | 0.029 | 0.822 | 0.960 | **0.870** |
| | | ExAUC | 0.516 | 0.029 | 0.518 | 0.013 | 0.366 | 0.028 | **0.845** | **0.963** | 0.858 |
| | | SqAUC | 0.407 | 0.028 | 0.548 | 0.012 | 0.327 | 0.031 | 0.811 | 0.933 | 0.867 |
| | | NWAUC | 0.565 | 0.030 | 0.574 | 0.017 | 0.396 | 0.027 | 0.786 | 0.885 | 0.827 |
| | | PAUC-exp | 0.549 | 0.029 | 0.508 | 0.015 | 0.354 | 0.028 | 0.650 | 0.801 | 0.736 |
| | | PAUC-poly | 0.526 | 0.028 | 0.470 | 0.015 | 0.354 | 0.029 | 0.661 | 0.812 | 0.742 |
| | | PAUCI | 0.516 | 0.027 | 0.520 | 0.015 | 0.382 | 0.028 | 0.704 | 0.847 | 0.734 |
| | | CS-hinge | 0.566 | 0.026 | 0.633 | **0.010** | 0.377 | 0.022 | 0.675 | 0.782 | 0.762 |
| | | AdaCOS | 0.576 | 0.025 | 0.559 | 0.014 | 0.391 | 0.023 | 0.758 | 0.873 | 0.742 |
| | | ECL | 0.589 | 0.026 | 0.561 | 0.014 | 0.388 | 0.020 | 0.694 | 0.918 | 0.762 |
| | Our method | WAUC-Gau | **0.679** | 0.024 | 0.660 | 0.012 | 0.467 | 0.015 | 0.787 | 0.934 | 0.843 |
| | | WAUC-Log | 0.653 | **0.023** | **0.674** | **0.011** | **0.468** | 0.014 | 0.820 | 0.958 | 0.869 |
| CIFAR-100-LT | Competitors | BCE | 0.556 | 0.022 | 0.463 | 0.012 | 0.512 | 0.019 | 0.912 | 0.957 | 0.806 |
| | | ExAUC | 0.522 | 0.019 | 0.502 | 0.011 | 0.506 | 0.017 | **0.933** | **0.967** | 0.833 |
| | | SqAUC | 0.483 | 0.024 | 0.367 | 0.015 | 0.474 | 0.018 | 0.889 | 0.955 | 0.855 |
| | | NWAUC | 0.654 | 0.025 | 0.511 | 0.016 | 0.631 | 0.019 | 0.867 | 0.925 | 0.807 |
| | | PAUC-exp | 0.464 | 0.020 | 0.282 | 0.014 | 0.469 | 0.016 | 0.826 | 0.811 | 0.787 |
| | | PAUC-poly | 0.461 | 0.022 | 0.262 | 0.017 | 0.473 | 0.017 | 0.828 | 0.887 | 0.791 |
| | | PAUCI | 0.549 | 0.018 | 0.439 | 0.016 | 0.514 | 0.018 | 0.812 | 0.843 | 0.822 |
| | | CS-hinge | 0.523 | 0.017 | 0.457 | 0.010 | 0.515 | 0.014 | 0.734 | 0.910 | 0.716 |
| | | AdaCOS | 0.590 | 0.018 | 0.474 | 0.011 | 0.587 | 0.016 | 0.769 | 0.919 | 0.727 |
| | | ECL | 0.583 | 0.017 | 0.497 | 0.009 | 0.595 | 0.015 | 0.863 | 0.939 | 0.794 |
| | Our method | WAUC-Gau | **0.745** | 0.015 | **0.589** | 0.005 | 0.728 | 0.013 | 0.842 | 0.928 | 0.745 |
| | | WAUC-Log | 0.719 | **0.012** | 0.560 | **0.003** | **0.745** | **0.010** | 0.906 | 0.960 | **0.875** |

Table 3: Performance comparisons on benchmark datasets in real world cost-sensitive problem. Profit means represents the money earned by the model over the entire trading period.

| Methods | BCE | ExAUC | SqAUC | NWAUC | PAUC-exp | PAUC-poly | PAUCI | CS-hinge | AdaCOS | ECL | WAUC-Gau | WAUC-Log |
|---|---|---|---|---|---|---|---|---|---|---|---|---|
| $\widehat{WAUC}\uparrow$ | 0.5427 | $0.594\pm.003$ | $0.508\pm.005$ | $0.562\pm.002$ | $0.576\pm.004$ | $0.481\pm.005$ | $0.529\pm.006$ | $0.592\pm.002$ | $0.6527\pm.005$ | $0.625\pm.004$ | $\mathbf{0.698}\pm.002$ | $\underline{0.675}\pm.001$ |
| $\widehat{\mathcal{L}}_{COST}\downarrow$ | $0.254\pm.004$ | $0.269\pm.005$ | $0.246\pm.003$ | $0.251\pm.002$ | $0.270\pm.004$ | $0.246\pm.004$ | $0.243\pm.001$ | $0.237\pm.006$ | $0.229\pm.004$ | $0.226\pm.007$ | $\mathbf{0.209}\pm.003$ | $\underline{0.213}\pm.002$ |
| $\widehat{AUC}\uparrow$ | $0.528\pm.005$ | $\mathbf{0.539}\pm.004$ | $0.526\pm.005$ | $0.520\pm.004$ | $0.529\pm.002$ | $0.519\pm.005$ | $0.510\pm.003$ | $0.522\pm.005$ | $0.5246\pm.002$ | $\underline{0.530}\pm.003$ | $0.526\pm.002$ | $0.5235\pm.003$ |
| Profit $\uparrow$ | $4955\pm20.14$ | $5468\pm17.90$ | $5183\pm30.91$ | $5395\pm22.48$ | $5418\pm14.06$ | $4862\pm28.04$ | $4963\pm15.09$ | $5583\pm30.05$ | $5839\pm34.92$ | $5764\pm25.09$ | $\mathbf{6526}\pm15.98$ | $\underline{6308}\pm16.09$ |

## 7.2 Overall Performance

In Tab. 2 and Tab. 3, we collect all the methods' performance on test sets of three types of datasets. For cost distribution of $c$, we sample some data from a normal distribution $\mathcal{N}(0.5, 1)$ to construct a dataset $S_c$ (we clip all data to $[0, 1]$). We also conduct numerous experiments for other types of distribution of $c$, and please see Appendix B for the details. From the results, we make the following observations:

(1) For $\widehat{WAUC}$ and $\widehat{\mathcal{L}}_{COST}$ metric, Our proposed algorithm achieves superior performance in most benchmark datasets compared to other methods. This demonstrates that our proposed WAUC cost-sensitive learning can extend the ROC curve into the cost space. Models trained with our proposed bilevel optimization formulation can enjoy high WAUC and cost-related metrics.

(2) AUC and cost-related metrics are inconsistent. From the high-performing heatmap of Tab. 2, it can be noticed that $\widehat{\mathcal{L}}_{COST}$ and $\widehat{AUC}$ have two completely different highlight regions. This indicates that the assumption of uniform distribution of AUC does not match the realistic scenario.

(3) We also find that AUC-related and traditional classification algorithms do not perform well in cost-sensitive problems. This means that if we first train the model with the classification algorithm, subsequently using the cost function to solve for the optimal threshold for decision does not work well. Meanwhile, the algorithm that can learn from scratch has better scalability. Two-stage decision method Therefore, designing one-stage algorithms for WAUC cost-sensitive learning is necessary.

### 7.3 Sensitivity Analysis

In this subsection, we show the sensitivity of $\beta$, $T$, and bandwidth on test data.

**Effect of $\beta$.** In Fig. 3 (a) and (d), we observe that for both WAUC and cost metrics, when $\beta$ closes to 7, the model will have the largest performance improvement and the lowest variance. This can be explained in two ways: (1) When the $\beta$ is too small, the error between the $\sigma(x)$ and the 0-1 loss function is large, resulting in a large approximation error between the $\widehat{\text{WAUC}}$ and the WAUC. (2) When the $\beta$ is too large, the gradient also tends to be 0. Therefore, choosing a beta value that trades off the approximation error and the gradient is essential.

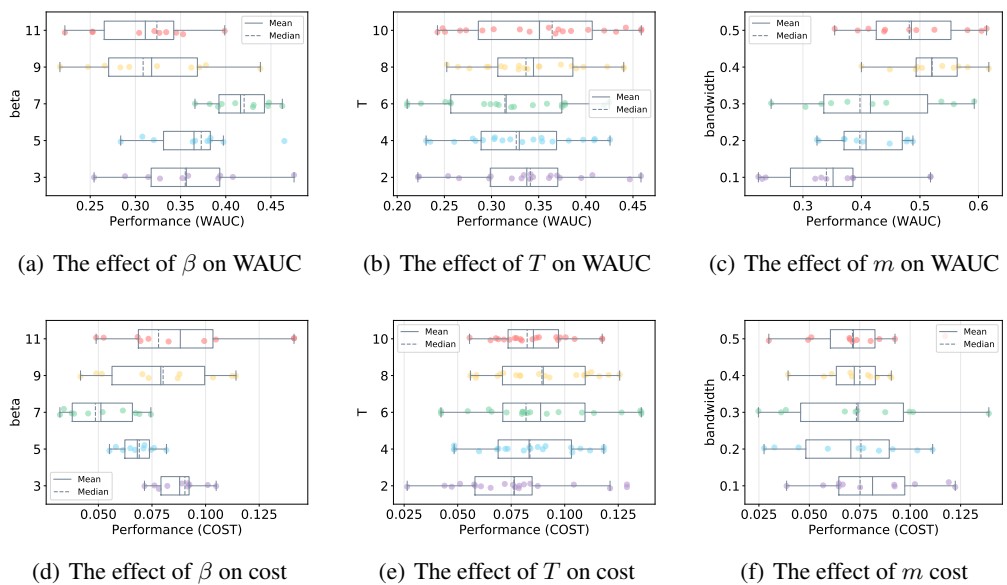

(a) The effect of $\beta$ on WAUC  (b) The effect of $T$ on WAUC  (c) The effect of $m$ on WAUC

(d) The effect of $\beta$ on cost  (e) The effect of $T$ on cost  (f) The effect of $m$ cost

Figure 3: Sensitivity analysis on test data where WAUC and cost for WAUC-Gau with respect to $\beta$, $T$, and bandwidth. The other two variables are fixed for each box in the plots, and the scattered points along the box show the variation.

**Effect of $T$.** As we mentioned in the Section 6.1, choosing a smaller $T$ can effectively improve the performance of the model. However, as shown in Fig. 3(e), a larger $T$ value can reduce the variance. Hence, as set in our experiments, $T = 3$ is a good choice to ensure the average performance and variance of the model.

**Effect of $m$.** From Fig. 3(c), we find that the kernel's bandwidth strongly influences the model's performance. The model's bandwidth and performance are almost proportional; the closer the bandwidth is to [0.4, 0.5], the better the effect; otherwise, the effect is worse. This indicates that our proposed method is sensitive to the bandwidth parameter, which also compounds the bandwidth characteristics in the KDE method.

## 8 Conclusion

This paper focuses on extending the traditional AUC metric to associate with misclassification costs. Restricted by the assumption of cost distribution, existing settings could not describe the model's performance in the complicated cost-sensitive scenario. To address this problem, we propose a novel setting that treats the cost as sampled data. We employ the WAUC metric and propose a novel estimator to approximate it. With the help of threshold weighting, we establish the correspondence between WAUC and the cost function. To describe this connection, we present a bilevel optimization formulation to couple them, where the inner-level problem provides a threshold optimization process, and the outer-level minimizes the WAUC loss based on the inner thresholds. This paradigm ensures that the WAUC can always be optimized at the optimal threshold value based on the complicated cost distribution in reality. Moreover, we propose a stochastic algorithm to optimize this formulation. We prove that our algorithm enjoys the same convergence rate as standard SGD. Finally, numerous

experiments have shown that our method can extend AUC to cost-sensitive scenarios with significant performance.

## Acknowledgements

This work was supported in part by the National Key R&D Program of China under Grant 2018AAA0102000, in part by National Natural Science Foundation of China: 62236008, U21B2038, U2001202, 61931008, 62122075, 61976202, and 62206264, in part by the Fundamental Research Funds for the Central Universities, in part by Youth Innovation Promotion Association CAS, in part by the Strategic Priority Research Program of Chinese Academy of Sciences (Grant No. XDB28000000) and in part by the Innovation Funding of ICT, CAS under Grant No. E000000.

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
