# A  Related Work

**ROC curve & cost-sensitive learning.** ROC curve and cost curve are two statistical tools frequently used in machine learning applications. Over the past two decades, some studies have explored their relationship and achieved significant success. [16] found that AUC implicitly uses a threshold weighting function corresponding to a cost weighting function. When these weighting functions output constant, we can infer that the cost function is a linear transformation of AUC. [21] shown that utilizing a natural threshold choice method can transfer ROC curves to cost space. [20] proposed a unified view of performance metrics. With the help of ROC convex hull [9], they give a clear interpretation of the threshold choice of the ROC curve. However, all of the above studies are based on the assumption of a uniform distribution in costs and thresholds. Moreover, their method can not be applied to end-to-end learning.

**AUC.** Since AUC offers some excellent properties in classification, it has become one of the standard performance metrics for binary imbalanced learning [1, 31, 43]. A partial list of the related studies includes [13, 7, 42, 22, 35, 12, 36, 47]. In deep age, there are some studies [27, 15, 48] focused on applying the AUC metric to deep end-to-end learning.

**PAUC.** The concept of PAUC was first introduced by [31], mainly used in disease diagnosis and biology. Prior to the deep learning era, earlier studies focused on optimizing PAUC in negative cost-sensitive scenarios. A partial list of the related studies includes [34, 33, 45, 50, 46, 38]. However, both AUC nor PAUC optimization does not consider the relationship between the ROC curve and cost space, which is hard to be applied in reality.

**WAUC.** The idea of weighting thresholds in AUC is first described by [40]. [25] constructed a framework for ROC analysis that incorporates the specificity distribution (*e.g.*, normal distribution). [30] proved exponential bounds on the estimation error of their proposed WAUC estimator and given conditions of the weight function. However, the weighting functions of these works are pre-selected, which are problem-independent, do not relate to the cost, and are too far from the practical application.

**Bilevel optimization.** Bilevel optimization is a classical algorithm for operations research. This formulation focuses on coupled optimization problems, where the inner optimization problem is the constraint of the outer optimization [3, 39, 6]. Recently, many studies have focused on designing gradient-based first-order algorithms to solve bilevel optimization problems [37, 11, 28, 5]. Most of them assume that the inner optimization problem has good properties, *e.g.*, strong convexity.

# B  Experiment details

## B.1  Main Idea of Experiments

Our experiments mainly explore the following three problems:

- Our method versus the recent SOTA cost-sensitive learning algorithm. Direct optimization of cost function leads to the model is sensitive to the class distribution, while AUC has the advantage of being robust to the class distribution. Compared with previous cost-sensitive learning methods, we combine the advantages of AUC and cost metrics, allowing the model to enjoy a higher WAUC value while minimizing the cost. Models trained with our method can be applied to cost-sensitive decision problems (e.g., financial market prediction, higher WAUC to guarantee decision profit)

- Traditional AUC is inconsistent with the cost-related metrics and cannot be used in cost-sensitive learning scenarios. From the experimental results in our paper, we can see that most AUC optimization methods do not minimize the misclassification cost. This indicates that in practical applications, using AUC optimization may maximize revenue without considering the cost. Ultimately, the misclassification cost of the decision is not acceptable.

- Our method versus the model trained with AUC/PAUC/CE first and then solves the optimal threshold with $\hat{\mathcal{L}}_{COST}$ for a decision. We want to prove that model trained with AUC/PAUC/CE first and then getting the posterior threshold performs poorly in cost-sensitive learning. Hence, developing a novel one-stage method to address this problem is necessary.

### B.2 Dataset Details

**Binary CIFAR-10-Long-Tail Dataset.** The CIFAR-10 dataset [23] consists of 60,000 images divided into ten categories. By choosing one superclass as positive and the other as negative, we construct a long-tail binary version of CIFAR-10. For scalability, we generate three subsets of CIFAR-10 which are composed of the different superclasses, including 1) birds, 2) automobiles, and 3) cats.

**Binary CIFAR-100-Long-Tail Dataset.** Different from CIFAR-10, We choose a set of classes in CIFAR-100 [23] as positive and the rest of the classes as negative. Similarly, we construct three subsets and the positive set containing 1) insects, 2) vegetables and fruits, and 3) large omnivores and herbivores.

**Jane Street Market Prediction.** In reality, some applications like financial markets prediction [14] involve investment issues. Every investment has a cost involved. Developing trading strategies to identify and take advantage of inefficiencies is challenging. We adopt the actual financial markets data [14] to test all methods on real cost situation.

### B.3 Implementation Details

We conducted all experiments on a Ubuntu 16.04.1 server equipped with an Intel(R) Xeon(R) Silver 4110 CPU and four RTX 3090 GPUs. We implement all algorithms code in `Python 3.8` and `pytorch 1.8.2` environment. For the fairness of the experiment, we adopt the ResNet-18 model as the backbone of all competitors. The model's output will be scaled into $[0, 1]$ with a `Sigmoid` function. We set the batch size as $256$ ($64$ per GPU) and epochs as $50$. We employ `torch.optim.SGD` as the basic optimizer and `torch.nn.DataParallel` as tools for parallel computing. We set $n_c$ as $50$.

### B.4 Parameter Tuning

We tune the learning rate of all methods in the range $[10^{-2}, 10^{-5}]$ and the weight decay in $[10^{-3}, 10^{-5}]$. Following the original paper, the warm-up steps for PAUC-poly and PAUC-exp is tuned in $[3, 20]$. Specifically, control parameter $\gamma$ in PAUC-poly is searched in $\{0.03, 0.05, 0.08, 0.1, 1, 3, 5\}$. For PAUC-exp, $\gamma$ is searched in $[8, 30]$. For PAUCI, stepsize related parameter $k$ is tuned in $[1, 10]$, $\nu$, $\lambda$, $c_1$, $c_2$ are searched in $[0, 1]$. $m$, $\kappa$, $\omega$ is tuned in $[10, 100]$, $[2, 6]$, $[0, 4]$ respectively. For our method, we set inner loop iterations $T = 3$, tune bandwidth $m$ in $[0.1, 0.5]$, smooth parameter $\beta$ in $[1, 10]$ and $M$, $M'$, $\kappa$ in $[32, 64, 128, 256, 512, 1024]$.

### B.5 Competitors

We implement two types of our methods, including Gaussian kernel $\frac{1}{\sqrt{2\pi}} \exp\left(-\frac{1}{2}x^2\right)$ and Logistics kernel $\frac{1}{e^x + 2 + e^{-x}}$ respectively. We denote them as WAUC-Gau and WAUC-Log. For other competitors, we classify them into three categories

**(1)** Common methods for binary classification problems, including the class balanced CE loss (BCE) [8]; the Exp loss of AUC (ExAUC); the square loss of AUC (SqAUC); the naive WAUC estimator (NWAUC).

**(2)** Recent SOTA methods of PAUC, including approximated PAUC estimator PAUC-poly [45] (poly calibrated weighting function) and PAUC-exp [45] (exp calibrated weighting function); asymptotically unbiased instance-wise regularized PAUC optimization [38] which is denoted as PAUCI.

**(3)** The cost-sensitive learning algorithms, including cost-sensitive classification with rejection which is denoted as CS-hinge [4], Bayes risk estimator for cost-sensitive classification [2] which is denoted as AdaCOS, expected cost loss which is denoted as ECL.

### B.6 Experiment Results

We give the details of the dataset used in the experiment in the following table. All serial numbers and category names correspond to the information in the original dataset.

We also conduct experiments to study our method's performance under other categories of cost distributions. We sampled the cost from the truncated uniform distribution $U[0.5, 1]$ and beta

Table 4: Details of dataset.

| Dataset | Pos. Class ID | Pos. Class Name | # Pos | #Neg |
|---|---|---|---|---|
| CIFAR-10-LT-1 | 2 | birds | 1,508 | 8,907 |
| CIFAR-10-LT-2 | 1 | automobiles | 2,517 | 7,898 |
| CIFAR-10-LT-3 | 3 | birds | 904 | 9,511 |
| CIFAR-100-LT-1 | 6,7,14,18,24 | insects | 1,928 | 13,218 |
| CIFAR-100-LT-2 | 0,51,53,57,83 | fruits and vegatables | 885 | 14,261 |
| CIFAR-100-LT-3 | 15,19,21,32,38 | large omnivores herbivores | 1,172 | 13,974 |

distribution $beta(2,4)$. We then conduct experiments on all competitors in subsets of CIFAR-10-Long-Tail and CIFAR-100-Long-Tail. All configurations are consistent with the sec.7 except for the cost data set $S_c$.

Table 5: Performance comparisons on benchmark datasets with different metrics (cost sampled from uniform distribution). The first and second best results are highlighted with **bold text** and underline, respectively.

| dataset | type | methods | Subset1 $\widehat{WAUC}$↑ | $\widehat{\mathcal{L}}_{COST}$↓ | Subset2 $\widehat{WAUC}$↑ | $\widehat{\mathcal{L}}_{COST}$↓ | Subset3 $\widehat{WAUC}$↑ | $\widehat{\mathcal{L}}_{COST}$↓ | $\widehat{AUC}$↑ Subset1 | Subset2 | Subset3 |
|---|---|---|---|---|---|---|---|---|---|---|---|
| CIFAR-10-LT | Competitors | BCE | 0.524 | 0.078 | 0.633 | 0.054 | 0.387 | 0.046 | 0.776 | 0.908 | 0.810 |
| | | ExAUC | 0.472 | 0.076 | 0.709 | 0.098 | 0.456 | 0.050 | 0.787 | 0.736 | 0.856 |
| | | SqAUC | 0.438 | 0.075 | 0.642 | 0.082 | 0.316 | 0.044 | 0.802 | 0.941 | 0.852 |
| | | NWAUC | 0.573 | 0.063 | 0.714 | 0.072 | 0.469 | 0.041 | 0.777 | 0.882 | 0.809 |
| | | PAUC-exp | 0.424 | 0.087 | 0.697 | 0.122 | 0.352 | 0.057 | 0.754 | 0.810 | 0.814 |
| | | PAUC-poly | 0.431 | 0.087 | 0.629 | 0.094 | 0.328 | 0.054 | 0.763 | 0.760 | 0.766 |
| | | PAUCI | 0.439 | 0.069 | 0.651 | 0.085 | 0.339 | 0.050 | 0.786 | 0.768 | 0.847 |
| | | CS-hinge | 0.482 | 0.087 | 0.607 | 0.096 | 0.404 | 0.054 | 0.734 | 0.748 | 0.777 |
| | | AdaCOS | 0.545 | 0.071 | 0.631 | 0.086 | 0.365 | 0.048 | 0.747 | 0.903 | 0.798 |
| | | ECL | 0.469 | 0.087 | 0.749 | 0.110 | 0.307 | 0.042 | 0.697 | 0.665 | 0.842 |
| | Our method | WAUC-Gau | 0.642 | 0.065 | **0.781** | 0.040 | **0.511** | 0.042 | 0.790 | 0.942 | 0.833 |
| | | WAUC-Log | **0.668** | **0.062** | 0.752 | **0.037** | 0.509 | **0.040** | **0.819** | **0.952** | **0.864** |
| CIFAR-100-LT | Competitors | BCE | 0.555 | 0.049 | 0.382 | 0.017 | 0.340 | 0.045 | 0.868 | 0.947 | 0.765 |
| | | ExAUC | 0.601 | 0.069 | 0.517 | 0.011 | 0.428 | 0.039 | 0.710 | 0.956 | **0.906** |
| | | SqAUC | 0.570 | 0.042 | 0.438 | 0.015 | 0.304 | 0.038 | 0.905 | 0.957 | 0.845 |
| | | NWAUC | 0.490 | 0.051 | **0.552** | 0.016 | 0.384 | 0.038 | 0.860 | 0.954 | 0.842 |
| | | PAUC-exp | 0.466 | 0.079 | 0.416 | 0.040 | 0.319 | 0.051 | 0.811 | 0.499 | 0.776 |
| | | PAUC-poly | 0.424 | 0.067 | 0.429 | 0.028 | 0.319 | 0.051 | 0.751 | 0.836 | 0.784 |
| | | PAUCI | 0.468 | 0.050 | 0.471 | 0.017 | 0.327 | 0.045 | 0.852 | 0.890 | 0.747 |
| | | CS-hinge | 0.485 | 0.065 | 0.436 | 0.029 | 0.307 | 0.051 | 0.763 | 0.826 | 0.693 |
| | | AdaCOS | 0.599 | 0.065 | 0.412 | 0.022 | 0.330 | 0.049 | 0.762 | 0.921 | 0.693 |
| | | ECL | 0.503 | 0.078 | 0.504 | 0.031 | 0.318 | 0.051 | 0.845 | 0.787 | 0.722 |
| | Our method | WAUC-Gau | **0.687** | 0.047 | 0.547 | 0.016 | 0.409 | 0.043 | 0.869 | 0.932 | 0.763 |
| | | WAUC-Log | 0.663 | **0.035** | 0.534 | **0.010** | **0.479** | **0.026** | **0.906** | **0.960** | 0.891 |

Under the condition of beta cost distribution, the performance of all methods in Tab. 6 is similar to Tab. 2. We can get a similar analysis result. Under the condition of uniform cost distribution, in Tab. 5, WAUC will degenerate to AUC. Therefore the AUC-related optimization algorithm will perform well and the gap with our proposed method will become smaller. Since PAUC is a special version of AUC based on the assumption of truncated uniform distribution of costs, the related algorithm has a clear advantage. However, although these algorithms have improved performance on the $\widehat{WAUC}$ metric, the results on $\widehat{\mathcal{L}}_{COST}$ are not good. Our proposed WAUC cost-sensitive learning can enjoy high WAUC metrics and cost metrics on both different kinds of cost distributions.

For the calculation of the $\widehat{WAUC}$ metric needs to involve the solution of the optimal threshold $\hat{\tau}^*$. However, this optimization problem is non-convex and it is difficult to find the optimal solution quickly using existing optimization methods. Therefore, we propose an algorithm that can be solved within $O(n_+ + n_-)$ iterations to obtain the optimal threshold, with the following details

## B.7 Addditional Experiment

In Thm. 5.3, we propose a convex formulation which can approximate the $\widehat{\mathcal{L}}_{COST}$. According to the conditions of the penalty, we need $\kappa, M \to \infty$ and $M'^2 < M^2 \frac{6\kappa^2 e^{3\kappa}}{(e^\kappa + 1)^6}$ to ensure that Thm. 5.3 is approximately equivalent to $\widehat{\mathcal{L}}_{COST}$.

Table 6: Performance comparisons on benchmark datasets with different metrics (cost sampled from beta distribution). The first and second best results are highlighted with **bold text** and underline, respectively.

| dataset | type | methods | Subset1 WAÛC ↑ | Subset1 $\widehat{\mathcal{L}}_{COST}$ ↓ | Subset2 WAÛC ↑ | Subset2 $\widehat{\mathcal{L}}_{COST}$ ↓ | Subset3 WAÛC ↑ | Subset3 $\widehat{\mathcal{L}}_{COST}$ ↓ | AÛC ↑ Subset1 | AÛC ↑ Subset2 | AÛC ↑ Subset3 |
|---|---|---|---|---|---|---|---|---|---|---|---|
| CIFAR-10-LT | Competitors | BCE | 0.316 | 0.033 | 0.474 | 0.032 | 0.412 | 0.024 | 0.821 | 0.915 | 0.806 |
| | | ExAUC | 0.389 | 0.035 | 0.474 | 0.037 | 0.491 | 0.021 | 0.841 | **0.967** | 0.867 |
| | | SqAUC | 0.270 | 0.033 | 0.495 | 0.039 | 0.425 | 0.022 | **0.858** | 0.933 | 0.855 |
| | | NWAUC | 0.326 | 0.034 | 0.538 | 0.033 | 0.462 | 0.023 | 0.817 | 0.922 | 0.826 |
| | | PAUC-exp | 0.071 | 0.033 | 0.490 | 0.059 | 0.476 | 0.027 | 0.783 | 0.715 | 0.740 |
| | | PAUC-poly | 0.137 | 0.035 | 0.477 | 0.045 | 0.445 | 0.026 | 0.746 | 0.830 | 0.703 |
| | | PAUCI | 0.185 | 0.038 | 0.526 | 0.038 | 0.476 | 0.022 | 0.796 | 0.844 | 0.787 |
| | | CS-hinge | 0.365 | 0.032 | 0.574 | 0.035 | 0.556 | 0.025 | 0.756 | 0.897 | 0.771 |
| | | AdaCOS | 0.347 | 0.038 | 0.567 | 0.036 | 0.516 | 0.024 | 0.803 | 0.900 | 0.808 |
| | | ECL | 0.303 | 0.034 | 0.584 | 0.030 | 0.537 | 0.022 | 0.847 | 0.925 | 0.856 |
| | Our method | WAUC-Gau | **0.413** | 0.031 | **0.640** | 0.030 | **0.610** | 0.020 | 0.815 | 0.943 | 0.830 |
| | | WAUC-Log | 0.393 | **0.029** | 0.578 | **0.024** | 0.575 | **0.019** | 0.852 | 0.959 | **0.869** |
| CIFAR-100-LT | Competitors | BCE | 0.547 | 0.025 | 0.423 | 0.016 | 0.190 | 0.027 | 0.869 | 0.930 | 0.757 |
| | | ExAUC | 0.646 | 0.023 | 0.459 | 0.015 | 0.184 | 0.021 | **0.929** | 0.948 | **0.900** |
| | | SqAUC | 0.460 | 0.023 | 0.396 | 0.009 | 0.157 | 0.025 | 0.892 | 0.949 | 0.854 |
| | | NWAUC | 0.652 | 0.024 | 0.494 | 0.009 | 0.151 | 0.026 | 0.889 | 0.941 | 0.819 |
| | | PAUC-exp | 0.566 | 0.034 | 0.464 | 0.015 | 0.184 | 0.028 | 0.800 | 0.835 | 0.793 |
| | | PAUC-poly | 0.269 | 0.029 | 0.410 | 0.012 | 0.167 | 0.028 | 0.788 | 0.885 | 0.740 |
| | | PAUCI | 0.530 | 0.024 | 0.449 | 0.007 | 0.193 | 0.025 | 0.827 | 0.895 | 0.705 |
| | | CS-hinge | 0.503 | 0.026 | 0.423 | 0.010 | 0.272 | 0.017 | 0.847 | 0.899 | 0.749 |
| | | AdaCOS | 0.605 | 0.027 | 0.452 | 0.011 | 0.244 | 0.017 | 0.852 | 0.915 | 0.696 |
| | | ECL | 0.503 | 0.025 | 0.424 | 0.010 | 0.220 | **0.016** | 0.872 | 0.930 | 0.799 |
| | Our method | WAUC-Gau | **0.747** | 0.022 | **0.658** | 0.008 | 0.275 | 0.023 | 0.869 | 0.925 | 0.757 |
| | | WAUC-Log | 0.647 | **0.018** | 0.645 | **0.005** | **0.290** | 0.017 | 0.911 | **0.961** | 0.865 |

---

**Algorithm 2** Algorithm for Solving the Optimal Threshold

0

**Input:** test data $S_+^t$ and $S_-^t$, cost dataset $S_c$. **Initialize:** parameters $\hat{\boldsymbol{\tau}}^* = \{0\}_{l=1}^{n_c}$
**for** $l = 0$ **to** $n_c$ **do**
 $\hat{\boldsymbol{\tau}}^*[l] = \arg\min_{\tau \in \{S_+^t, S_-^t\}} \widehat{\mathcal{L}}_{COST}$
**end for**

---

In this subsection, we study the optimal value gap between original $\widehat{\mathcal{L}}_{COST}$ problem and Thm. 5.3. We optimize the $\widehat{\mathcal{L}}_{COST}$ and Eq. (12) in Cifar-10-Long-Tail training set. We set cost distribution as Uniform and $\pi = 0.5$. We calculate the mean optimal value ($p^* = \min \widehat{\mathcal{L}}_{COST}$, $d^* = \min$ Eq. (12)) of them over the cost distribution. Finally, we calculate the error between them ($(p^* - d^*)^2$) and list the results in Tab. 7.

Table 7: Optimal value gap between original $\widehat{\mathcal{L}}_{COST}$ optimization problem and Thm. 5.3

| | $M = 64$ $\kappa = 64$ | $M = 128$ $\kappa = 128$ | $M = 256$ $\kappa = 256$ | $M = 512$ $\kappa = 512$ | $M = 1024$ $\kappa = 1024$ | $M = 2048$ $\kappa = 2048$ |
|---|---|---|---|---|---|---|
| $M' = 32$ | 0.058 | 0.050 | 0.048 | 0.045 | 0.042 | 0.041 |
| $M' = 64$ | 0.049 | 0.042 | 0.039 | 0.036 | 0.034 | 0.032 |
| $M' = 128$ | 0.038 | 0.036 | 0.032 | 0.029 | 0.027 | 0.025 |
| $M' = 256$ | 0.025 | 0.023 | 0.020 | 0.018 | 0.016 | 0.015 |
| $M' = 512$ | 0.018 | 0.015 | 0.013 | 0.011 | 0.009 | 0.008 |
| $M' = 1024$ | 0.012 | 0.010 | 0.008 | 0.007 | 0.007 | 0.005 |

In Tab. 7, we find that when $\kappa$, $M$, $M'$ grows, the optimal value gap will increase quickly. Moreover, when $\kappa = M = 64$ and $M' = 32$, the error is small enough. Hence, in real-world application, the approximation error and effect of hyparameters is acceptable.

## C  Proofs for Section 4

### C.1  KDE Definition

Here, we give the notation and definition of KDE.

**Definition C.1** (**Kernel density estimation** [49]). Denote $x$ as a random variable with probability density function $f_x$. Given a dataset $S = \{x_i\}_{i=1}^{n_x}$ and threshold $\tau$, we denote

$$\mathcal{K}(S, \tau) = \frac{1}{|S|m} \sum_{x_i \in S} K\left(\frac{x_i - \tau}{m}\right), \tag{17}$$

as an estimation of $f_x$, where $m$ is bandwidth. The non-negative real-valued integrable function $K$ satisfies

$$\textbf{(1)} \int_{-\infty}^{\infty} K(x)dx = 1, \ \textbf{(2)} \ K(x) = K(-x). \tag{18}$$

## C.2 Proof of Proposition 5.1

**Restatement of Proposition 5.1.** *Denote $K(x)$ be statistics kernel with bandwidth $m$ and $S_-^{\boldsymbol{w}} = \{s(\boldsymbol{w}, \boldsymbol{x}_j^-)\}_{j=1}^{n_-}$. With Lemma 5.2, we have the approximate estimator and loss function for WAUC:*

$$\widehat{\text{WAUC}} = \int_{\infty}^{-\infty} \text{TPR}_s(\tau)\mathcal{K}(S_-^{\boldsymbol{w}}, \tau)p(\tau)d\tau, \quad \widehat{\mathcal{L}}_{\text{WAUC}}(\boldsymbol{w}, \boldsymbol{\tau}) = \frac{1}{n_\tau}\sum_{l=1}^{n_\tau} \hat{h}(\boldsymbol{w}, \tau_l) \tag{19}$$

*where $\boldsymbol{\tau} = \{\tau_l\}_{l=1}^{n_\tau}$ and the point loss $\hat{h}$ is defined by*

$$\hat{h}(\boldsymbol{w}, \tau) = 1 - \frac{1}{n_+n_-} \sum_{i=1}^{n_+}\sum_{j=1}^{n_-} \sigma(s(\boldsymbol{w}, \boldsymbol{x}_i^+) - \tau_l) \cdot K((s(\boldsymbol{w}, \boldsymbol{x}_j^-) - \tau_l)/m)/m. \tag{20}$$

*$\sigma(x) = 1/(1 + \exp(-\beta x))$, $\beta$ is smooth parameter and we have $\sigma(x) \xrightarrow{\beta \to \infty} \mathbb{I}_x$.*

*Proof.*

$$\begin{aligned}
\widehat{\text{WAUC}} &= \int_{\infty}^{-\infty} \text{TPR}(\tau)\mathcal{K}(S_-, \tau)p(\tau)d\tau \\
&= \mathbb{E}_\tau\left[\mathbb{P}_{\boldsymbol{x}^+}[s(\boldsymbol{w}, \boldsymbol{x}^+) > \tau] \cdot \mathbb{E}_{\boldsymbol{x}^-}\left[\frac{1}{m}K\left(\frac{s(\boldsymbol{w}, \boldsymbol{x}^-) - \tau}{m}\right)\right]\right] \\
&= \mathbb{E}_\tau\left[\mathbb{E}_{\boldsymbol{x}^+}[\mathbb{I}_{s(\boldsymbol{w}, \boldsymbol{x}^+) > \tau}] \cdot \mathbb{E}_{\boldsymbol{x}^-}\left[\frac{1}{m}K\left(\frac{s(\boldsymbol{w}, \boldsymbol{x}^-) - \tau}{m}\right)\right]\right] \\
&= \mathbb{E}_{\tau, \boldsymbol{x}^+}\left[\mathbb{I}_{s(\boldsymbol{w}, \boldsymbol{x}^+) > \tau} \cdot \mathbb{E}_{\boldsymbol{x}^-}\left[\frac{1}{m}K\left(\frac{s(\boldsymbol{w}, \boldsymbol{x}^-) - \tau}{m}\right)\right]\right] \\
&= \mathbb{E}_{\tau, \boldsymbol{x}^+, \boldsymbol{x}^-}\left[\mathbb{I}_{s(\boldsymbol{w}, \boldsymbol{x}^+) > \tau} \cdot \left[\frac{1}{m}K\left(\frac{s(\boldsymbol{w}, \boldsymbol{x}^-) - \tau}{m}\right)\right]\right].
\end{aligned} \tag{21}$$

Replacing the $\mathbb{I}_{(\cdot)}$ function with $\sigma(x) = 1/(1 + \exp(-\beta x))$ and change it to empirical formulation.

$$\widehat{\mathbb{E}}_{\boldsymbol{x}^+ \sim S_+, \boldsymbol{x}^- \sim S_-, \tau \sim \boldsymbol{\tau}}\left[\sigma(s(\boldsymbol{w}, \boldsymbol{x}^+) - \tau) \cdot \left[\frac{1}{m}K\left(\frac{s(\boldsymbol{w}, \boldsymbol{x}^-) - \tau}{m}\right)\right]\right] \tag{22}$$

Since we want to maximize the WAUC metric, we employ $1 - \widehat{\text{WAUC}}$ as loss function, then we have

$$(\text{Pop.}) \ \mathcal{L}_{\text{WAUC}} = \mathop{\mathbb{E}}_{\boldsymbol{x}^+ \sim \mathcal{D}_\mathcal{P}, \boldsymbol{x}^- \sim \mathcal{D}_\mathcal{N}}\left[1 - \frac{1}{n_\tau}\sum_{l=1}^{n_\tau} \sigma(s(\boldsymbol{w}, \boldsymbol{x}^+) - \tau_l) \cdot \frac{1}{m}K\left(\frac{s(\boldsymbol{w}, \boldsymbol{x}^-) - \tau_l}{m}\right)\right]$$

$$(\text{Emp.}) \ \widehat{\mathcal{L}}_{\text{WAUC}} = \mathop{\widehat{\mathbb{E}}}_{\boldsymbol{x}^+ \sim S_+, \boldsymbol{x}^- \sim S_-}\left[1 - \frac{1}{n_\tau}\sum_{l=1}^{n_\tau} \sigma(s(\boldsymbol{w}, \boldsymbol{x}^+) - \tau_l) \cdot \frac{1}{m}K\left(\frac{s(\boldsymbol{w}, \boldsymbol{x}^-) - \tau_l}{m}\right)\right]. \tag{23}$$

Reformulating the $\widehat{\mathcal{L}}_{\text{WAUC}}$, we have:

$$\widehat{\mathcal{L}}_{\text{WAUC}}(\boldsymbol{w}, \boldsymbol{\tau}) = \frac{1}{n_\tau}\sum_{l=1}^{n_\tau} \hat{h}(\boldsymbol{w}, \tau_l) \tag{24}$$

where

$$\hat{h}(\boldsymbol{w},\tau) = 1 - \frac{1}{n_+ n_-} \sum_{i=1}^{n_+} \sum_{j=1}^{n_-} \sigma(s(\boldsymbol{w},\boldsymbol{x}_i^+) - \tau_l) \cdot K((s(\boldsymbol{w},\boldsymbol{x}_j^-) - \tau_l)/m)/m. \qquad (25)$$

$\sigma(x) = 1/(1 + \exp(-\beta x))$, $\beta$ is smooth parameter and we have $\sigma(x) \overset{\beta \to \infty}{\Longrightarrow} \mathbb{I}_x$ to approximate $\widehat{\mathrm{TPR}}$ and $\widehat{\mathrm{FPR}}$. $\qquad \square$

## C.3 Proof of Lemma 5.2

**Restatement of Lemma 5.2.** *Given a scoring function $s$, if $\tau$ is known, when the number of instances is large enough, $\widehat{\mathrm{WAUC}}$ almost surely converges to WAUC.*

$$\lim_{n_- \to \infty} |\widehat{\mathrm{WAUC}} - \mathrm{WAUC}| \overset{a.s.}{\longrightarrow} 0. \qquad (26)$$

*Proof.*

$$\lim_{n_- \to \infty} |\widehat{\mathrm{WAUC}} - \mathrm{WAUC}|$$

$$= \lim_{n_- \to \infty} \left| \hat{\mathbb{E}}_{\tau,\boldsymbol{x}^+} \left[ \sigma(s(\boldsymbol{w},\boldsymbol{x}^+) - \tau) \cdot \hat{\mathbb{E}}_{\boldsymbol{x}^-} \left[ \frac{1}{m} K\left( \frac{s(\boldsymbol{w},\boldsymbol{x}^-) - \tau}{m} \right) \right] \right] - \mathbb{E}_{\tau,\boldsymbol{x}^+} \left[ \mathbb{I}_{s(\boldsymbol{w},\boldsymbol{x}^+)>\tau} \cdot \mathrm{FPR}'_s(\tau) \right] \right|$$

$$\leq \lim_{n_- \to \infty} \sup_{\tau,\boldsymbol{x}^+} \left| \sigma(s(\boldsymbol{w},\boldsymbol{x}^+) - \tau) \cdot \hat{\mathbb{E}}_{\boldsymbol{x}^-} \left[ \frac{1}{m} K\left( \frac{s(\boldsymbol{w},\boldsymbol{x}^-) - \tau}{m} \right) \right] - \mathbb{I}_{s(\boldsymbol{w},\boldsymbol{x}^+)>\tau} \cdot \mathrm{FPR}'_s(\tau) \right|$$

$$\leq \lim_{n_- \to \infty} \sup_{\tau} \max \left\{ \underbrace{\left| \sigma(\delta)\hat{\mathbb{E}}_{\boldsymbol{x}^-} \left[ \frac{1}{m} K\left( \frac{s(\boldsymbol{w},\boldsymbol{x}^-) - \tau}{m} \right) \right] - \mathrm{FPR}'(\tau) \right|}_{\sigma(\delta) \to 1 \text{ when } \beta \to \infty}, \underbrace{\sigma(-\delta)\hat{\mathbb{E}}_{\boldsymbol{x}^-} \left[ \frac{1}{m} K\left( \frac{s(\boldsymbol{w},\boldsymbol{x}^-) - \tau}{m} \right) \right]}_{\sigma(-\delta) \to 0 \text{ when } \beta \to \infty} \right\}$$

$$\leq \lim_{n_- \to \infty} \sup_{\tau} \left| \hat{\mathbb{E}}_{\boldsymbol{x}^-} \left[ \frac{1}{m} K\left( \frac{s(\boldsymbol{w},\boldsymbol{x}^-) - \tau}{m} \right) \right] - \mathrm{FPR}'_s(\tau) \right|$$

$$\overset{(a)}{\leq} \lim_{n_- \to \infty} \sup_{\tau} \underbrace{\left| \hat{\mathbb{E}}_{\boldsymbol{x}^-} \left[ \frac{1}{m} K\left( \frac{s(\boldsymbol{w},\boldsymbol{x}^-) - \tau}{m} \right) \right] - \mathbb{E}\left[ \hat{\mathbb{E}}_{\boldsymbol{x}^-} \left[ \frac{1}{m} K\left( \frac{s(\boldsymbol{w},\boldsymbol{x}^-) - \tau}{m} \right) \right] \right] \right|}_{(c) \text{ Kernel density function consistency lemma}}$$

$$+ \underbrace{\lim_{n_- \to \infty} \sup_{\tau} \left| \mathbb{E}\left[ \hat{\mathbb{E}}_{\boldsymbol{x}^-} \left[ \frac{1}{m} K\left( \frac{s(\boldsymbol{w},\boldsymbol{x}^-) - \tau}{m} \right) \right] \right] - \mathrm{FPR}'_s(\tau) \right|}_{(d) \text{ Law of large numbers}}$$

$$= 0$$

$$(27)$$

where $(a)$ comes from triangle inequality and $\delta > 0$. We assume that $\mathbb{E}_{\tau,\boldsymbol{x}^+}[\mathbb{I}_{s(\boldsymbol{w},\boldsymbol{x}^+)=\tau}] = 0$. For terms $(c)$ and $(d)$, please see the proof of [41]. $\qquad \square$

## C.4 Proof of Theorem 5.3

**Restatement of Theorem 5.3.** *When we set $\kappa$, $M$ are large positive numbers and $M'^2 < M^2 \frac{6\kappa^2 e^{3\kappa}}{(e^\kappa+1)^6}$, then we have the approximated convex formulation for $\hat{\mathcal{L}}_{COST}$*

$$\min_{\tau,\boldsymbol{P} \in \mathbb{R}^{n_+}, \boldsymbol{N} \in \mathbb{R}^{n_-}} \hat{\mathcal{L}}_{eq}(\boldsymbol{w},\tau,c) := c \cdot \pi \cdot \left(1 - \frac{1}{n_+} \sum_{i=1}^{n_+} P_i\right) + (1-c) \cdot (1-\pi) \cdot \left(\frac{1}{n_-} \sum_{j=1}^{n_-} N_j\right)$$

$$+ \frac{1}{n_+} \sum_{i=1}^{n_+} M'\psi(s(\boldsymbol{w},\boldsymbol{x}_i^+) - \tau) - P_i(s(\boldsymbol{w},\boldsymbol{x}_i^+) - \tau)) + M\psi(P_i - 1) + M\psi(\tau - 1) \qquad (28)$$

$$+ \frac{1}{n_-} \sum_{j=1}^{n_-} M'\psi(s(\boldsymbol{w},\boldsymbol{x}_j^-) - \tau) - N_j(s(\boldsymbol{w},\boldsymbol{x}_j^-) - \tau)) + M\psi(N_j - 1) \quad 0 \leq \tau, P_i, N_j$$

where $\psi(x) = \log(1 + \exp(\kappa x))/\kappa$. $\widehat{\mathcal{L}}_{eq}$ in Eq.(12) is $\mu_g$-strongly convex w.r.t. $\boldsymbol{\tau}$. Eq.(12) has same solution as $\min_\tau \widehat{\mathcal{L}}_{COST}$ when the parameters satisfy the conditions of the penalty.

*Proof.* According to the definition of $\widehat{\mathcal{L}}_{COST}$, we have:

$$\min_\tau \; c \cdot \pi \cdot \left(1 - \frac{1}{n_+} \sum_{i=1}^{n_+} \mathbb{I}_{[s(\boldsymbol{w}, \boldsymbol{x}_i^+) - \tau]}\right) + (1 - c) \cdot (1 - \pi) \cdot \left(\frac{1}{n_-} \sum_{j=1}^{n_-} \mathbb{I}_{[s(\boldsymbol{w}, \boldsymbol{x}_j^-) - \tau]}\right) \quad (29)$$
$$s.t. \quad 0 \le \tau \le 1.$$

Then we have the equivalent formulation:

$$\min_{\tau, \boldsymbol{P}, \boldsymbol{N}} c \cdot \pi \cdot \left(1 - \frac{1}{n_+} \sum_{i=1}^{n_+} P_i\right) + (1 - c) \cdot (1 - \pi) \cdot \left(\frac{1}{n_-} \sum_{j=1}^{n_-} N_j\right)$$
$$s.t. \max(s(\boldsymbol{w}, \boldsymbol{x}_i^+) - \tau, 0) = P_i(s(\boldsymbol{w}, \boldsymbol{x}_i^+) - \tau) \quad (30)$$
$$\max(s(\boldsymbol{w}, \boldsymbol{x}_j^-) - \tau, 0) = N_j(s(\boldsymbol{w}, \boldsymbol{x}_j^-) - \tau)$$
$$0 \le \tau, P_i, N_j \le 1,$$

where $\boldsymbol{P} \in \mathbb{R}^{n_+}$, $\boldsymbol{N} \in \mathbb{R}^{n_-}$ and $P_i \in \boldsymbol{P}$, $N_j \in \boldsymbol{N}$. Since the equality constraint is hard to process, we convert it to inequality constraint (*e.g.*, $a = b \Leftrightarrow a <= b, a >= b$).

$$\min_{\tau, \boldsymbol{P}, \boldsymbol{N}} c \cdot \pi \cdot \left(1 - \frac{1}{n_+} \sum_{i=1}^{n_+} P_i\right) + (1 - c) \cdot (1 - \pi) \cdot \left(\frac{1}{n_-} \sum_{j=1}^{n_-} N_j\right)$$
$$s.t. \; \forall i \; \max(s(\boldsymbol{w}, \boldsymbol{x}_i^+) - \tau, 0) \ge P_i(s(\boldsymbol{w}, \boldsymbol{x}_i^+) - \tau)$$
$$\forall j \; \max(s(\boldsymbol{w}, \boldsymbol{x}_j^-) - \tau, 0) \ge N_j(s(\boldsymbol{w}, \boldsymbol{x}_j^-) - \tau) \quad (31)$$
$$\forall i \; \max(s(\boldsymbol{w}, \boldsymbol{x}_i^+) - \tau, 0) \le P_i(s(\boldsymbol{w}, \boldsymbol{x}_i^+) - \tau)$$
$$\forall j \; \max(s(\boldsymbol{w}, \boldsymbol{x}_j^-) - \tau, 0) \le N_j(s(\boldsymbol{w}, \boldsymbol{x}_j^-) - \tau)$$
$$\forall i, j \quad 0 \le \tau, P_i, N_j \le 1$$

Due to the fact that $\max(s(\boldsymbol{w}, \boldsymbol{x}_i^+) - \tau, 0) \ge P_i(s(\boldsymbol{w}, \boldsymbol{x}_i^+) - \tau)$ and $\max(s(\boldsymbol{w}, \boldsymbol{x}_j^-) - \tau, 0) \ge N_j(s(\boldsymbol{w}, \boldsymbol{x}_j^-) - \tau)$ is ground truth all the time. Hence, we have

$$\min_{\tau, \boldsymbol{P}, \boldsymbol{N}} c \cdot \pi \cdot \left(1 - \frac{1}{n_+} \sum_{i=1}^{n_+} P_i\right) + (1 - c) \cdot (1 - \pi) \cdot \left(\frac{1}{n_-} \sum_{j=1}^{n_-} N_j\right)$$
$$s.t. \; \forall i \; \max(s(\boldsymbol{w}, \boldsymbol{x}_i^+) - \tau, 0) \le P_i(s(\boldsymbol{w}, \boldsymbol{x}_i^+) - \tau) \quad (32)$$
$$\forall j \; \max(s(\boldsymbol{w}, \boldsymbol{x}_j^-) - \tau, 0) \le N_j(s(\boldsymbol{w}, \boldsymbol{x}_j^-) - \tau)$$
$$\forall ij \quad 0 \le \tau, P_i, N_j \le 1$$

Then we apply the penalty function method to convert the constraint optimization into approximated unconstrained optimization:

$$\min_{\tau, \boldsymbol{P}, \boldsymbol{N}} c \cdot \pi \cdot \left(1 - \frac{1}{n_+} \sum_{i=1}^{n_+} P_i\right) + (1 - c) \cdot (1 - \pi) \cdot \left(\frac{1}{n_-} \sum_{j=1}^{n_-} N_j\right) + M\psi(\tau - 1)$$
$$+ \frac{1}{n_+} \sum_{i=1}^{n_+} M'(\psi(s(\boldsymbol{w}, x_i^+) - \tau) - P_i(s(\boldsymbol{w}, x_i^+)) - \tau) + M\psi(P_i - 1)) \quad (33)$$
$$+ \frac{1}{n_-} \sum_{j=1}^{n_-} M'(\psi(s(\boldsymbol{w}, x_j^-) - \tau) - N_j(s(\boldsymbol{w}, x_j^-)) - \tau) + M\psi(N_j - 1))$$
$$\forall ij \quad 0 \le \tau, P_i, N_j$$

where $\psi(x) = \frac{\log(1+\exp(\kappa x))}{\kappa}$ is penalty function ($\psi(x) \overset{\kappa \to \infty}{\to} \max(x,0)$), $M$ and $M'$ denote positive number which are large enough. It's noticed that when $\kappa, M, M' \to \infty$, then Eq.(33) is equivalent to Eq.(32). Next, we will prove the strong convexity of $\tau$ in Eq.(33). Firstly, we give the hessian matrix of Eq.(33):

$$
H = M \begin{bmatrix} \frac{\kappa e^{\kappa(\tau-1)}}{(e^{\kappa(\tau-1)}+1)^2} \\ +\frac{1}{n_+}\sum_{i=1}^{n_+} \frac{\kappa e^{\kappa(P_i-\tau)}}{(e^{\kappa(P_i-\tau)}+1)^2} & M'/M & M'/M \\ +\frac{1}{n_-}\sum_{j=1}^{n_-} \frac{\kappa e^{\kappa(N_j-\tau)}}{(e^{\kappa(N_j-\tau)}+1)^2} \\ M'/M & \frac{1}{n_+}\sum_{i=1}^{n_+}\frac{\kappa e^{\kappa(P_i-1)}}{(e^{\kappa(P_i-1)}+1)^2} & 0 \\ M'/M & 0 & \frac{1}{n_-}\sum_{j=1}^{n_-}\frac{\kappa e^{\kappa(N_j-1)}}{(e^{\kappa(N_j-1)}+1)^2} \end{bmatrix}
$$
(34)

For computational simplicity, we define

$$
\begin{aligned}
x &= \frac{\kappa e^{\kappa(\tau-1)}}{(e^{\kappa(\tau-1)}+1)^2} + \frac{1}{n_+}\sum_{i=1}^{n_+}\frac{\kappa e^{\kappa(P_i-\tau)}}{(e^{\kappa(P_i-\tau)}+1)^2} + \frac{1}{n_-}\sum_{j=1}^{n_-}\frac{\kappa e^{\kappa(N_j-\tau)}}{(e^{\kappa(N_j-\tau)}+1)^2} \\
y &= \frac{1}{n_+}\sum_{i=1}^{n_+}\frac{\kappa e^{\kappa(P_i-1)}}{(e^{\kappa(P_i-1)}+1)^2} \\
z &= \frac{1}{n_-}\sum_{j=1}^{n_-}\frac{\kappa e^{\kappa(N_j-1)}}{(e^{\kappa(N_j-1)}+1)^2}
\end{aligned}
$$
(35)

Then we reformulate the hessian matrix

$$
H = M \begin{bmatrix} x & M'/M & M'/M \\ M'/M & y & 0 \\ M'/M & 0 & z \end{bmatrix}
$$
(36)

where

$$
x \in \left[\frac{3\kappa e^{\kappa}}{(e^{\kappa}+1)^2}, \frac{3\kappa}{4}\right], y, z \in \left[\frac{\kappa e^{\kappa}}{(e^{\kappa}+1)^2}, \frac{\kappa}{4}\right]
$$
(37)

We calculate the principal minor of the hessian matrix

$$
D_1 = x > 0
$$
(38)

$$
D_2 = xy - \frac{M'^2}{M^2} > 0 \Rightarrow M'^2 \le M^2\frac{3\kappa^2 e^{2\kappa}}{(e^{\kappa}+1)^4} < M^2 xy
$$
(39)

$$
D_3 = xyz - (y+z)\frac{M'^2}{M^2} > 0 \Rightarrow M'^2 \le M^2\frac{6\kappa^2 e^{3\kappa}}{(e^{\kappa}+1)^6} < \frac{M^2 xyz}{y+z}
$$
(40)

Hence, we find that if we have $M'^2 < \min\left(M^2\frac{3\kappa^2 e^{2\kappa}}{(e^{\kappa}+1)^4}, M^2\frac{6\kappa^2 e^{3\kappa}}{(e^{\kappa}+1)^6}\right)$, then we can ensure $\tau$ is strongly convex for Eq.(33). We define the approximated equivalent formulation

$$
\min_{\tau, \boldsymbol{P} \in \mathbb{R}^{n_+}, \boldsymbol{N} \in \mathbb{R}^{n_-}} \widehat{\mathcal{L}}_{eq}(\boldsymbol{w}, \tau, c) := c \cdot \pi \cdot (1 - \frac{1}{n_+}\sum_{i=1}^{n_+} P_i) + (1-c)\cdot(1-\pi)\cdot(\frac{1}{n_-}\sum_{j=1}^{n_-} N_j)
$$

$$
+ \frac{1}{n_+}\sum_{i=1}^{n_+} M'\psi(s(\boldsymbol{w}, \boldsymbol{x}_i^+) - \tau) - P_i(s(\boldsymbol{w}, \boldsymbol{x}_i^+) - \tau)) + M\psi(P_i - 1) + M\psi(\tau - 1)
$$
(41)

$$
+ \frac{1}{n_-}\sum_{j=1}^{n_-} M'\psi(s(\boldsymbol{w}, \boldsymbol{x}_j^-) - \tau) - N_j(s(\boldsymbol{w}, \boldsymbol{x}_j^-) - \tau)) + M\psi(N_j - 1) \; \forall ij \; 0 \le \tau, P_i, N_j
$$

Then we can calculate the strong convexity of $\tau$. According to the definition of strong convex

$$
\exists \mu > 0, \forall \tau \in [0,1], \boldsymbol{P} \in [0,1]^{n_+}, \boldsymbol{N} \in [0,1]^{n_-} \quad \nabla^2 \widehat{\mathcal{L}}_{eq} \succeq \mu_g I
$$
(42)

Assuming that $\mu_g > 0$, in order to satisfy the strong convexity, we need to ensure the positive definiteness of the Hessian matrix

$$
H = M \begin{bmatrix} x - \mu_g & M'/M & M'/M \\ M'/M & y - \mu_g & 0 \\ M'/M & 0 & z - \mu_g \end{bmatrix}
$$
(43)

where

$$x \in \left[\frac{3\kappa e^{\kappa}}{(e^{\kappa}+1)^2}, \frac{3\kappa}{4}\right], y, z \in \left[\frac{\kappa e^{\kappa}}{(e^{\kappa}+1)^2}, \frac{\kappa}{4}\right] \tag{44}$$

We calculate the principal minor of the Hessian matrix

$$D_1 = x - \mu_g > 0 \Rightarrow \mu_g < \frac{3\kappa e^{\kappa}}{(e^{\kappa}+1)^2} \tag{45}$$

$$D_2 = (x - \mu_g)(y - \mu_g) - \frac{M'^2}{M^2} > 0 \Rightarrow$$

$$\mu_g > \frac{\kappa\sqrt{\kappa^2 + \frac{3\kappa^2}{16}M'^2/M^2}}{2} \geq \frac{x + y + \sqrt{(x+y)^2 + 4xyM'^2/M^2}}{2} \tag{46}$$

$$D_3 = (x - \mu_g)(y - \mu_g)(z - \mu_g) - (y + z - 2\mu_g)\frac{M'^2}{M^2} > 0 \Rightarrow$$

$$\mu_g > \sqrt[3]{-\frac{q}{2} + \sqrt{\frac{q^2}{4} + \frac{p^3}{27}}} + \sqrt[3]{-\frac{q}{2} - \sqrt{\frac{q^2}{4} + \frac{p^3}{27}}} \tag{47}$$

where

$$p = -xy - xz - yz - \frac{(x+y+z)^2}{3} - \frac{M'(y+z)}{M}$$

$$q = xyz + \frac{2(x+y+z)^3}{27} - \frac{(9x+9y+9z)\left(-xy - xz - yz - \frac{M'(y+z)}{M}\right)}{27} \tag{48}$$

When $p = -\frac{23\kappa^2}{24} - \frac{M'\kappa}{2M}$ and $q = \frac{331k^3}{1728} - \frac{5k\left(-\frac{7k^2}{16} - \frac{M'k}{2M}\right)}{12}$, $\mu$ has a lower bound. Hence, we find that if we have

$$\max\left(\sqrt[3]{-\frac{q}{2} + \sqrt{\frac{q^2}{4} + \frac{p^3}{27}}} + \sqrt[3]{-\frac{q}{2} - \sqrt{\frac{q^2}{4} + \frac{p^3}{27}}}, \frac{\kappa\sqrt{\kappa^2 + \frac{3\kappa^2}{16}M'^2/M^2}}{2}\right)$$

$$< \mu_g < \min\left(\frac{\kappa e^{\kappa}}{(e^{\kappa}+1)^2}, \frac{3\kappa e^{\kappa}}{(e^{\kappa}+1)^2}\right) \tag{49}$$

then we can ensure $\tau$ is $\mu_g$-strongly convex for Eq.(33). For computational simplicity, we use the upper bound of $\mu = \frac{3M\kappa e^{\kappa}}{(e^{\kappa}+1)^2} \geq M \cdot \mu_g$. $\qquad\square$