# OpenReview forum: "Weighted ROC Curve in Cost Space: Extending AUC to Cost-Sensitive Learning"
_NeurIPS.cc/2023/Conference — NeurIPS 2023 poster_

### Official Review · Reviewer_Trb9 · 2023-07-06

**Soundness:** 3 good
**Presentation:** 3 good
**Contribution:** 3 good
**Rating:** 5
**Confidence:** 4

**Summary:**

This paper proposed more robust learning using combining the WAUC and cost learning. The authors claim that it can be vital to the shifts of cost function and covariate distribution. The algorithm is constructed using bi-level optimization with inner and outer parts. Experiments show high performance, especially in earning money in cost-sensitive situations.


**Strengths:**

The approach is somewhat similar to Bayesian, considering the randomness. The motivation is clear, and the algorithm is well-established. Furthermore, this paper is successful in providing the appropriate cases.

**Weaknesses:**

The presentation is not good. For example, the notation of $\hat{L}_{WAUC}$ is isolated in proposition 5.1. Also, there are many typos concerning the indexes in formulation and algorithm.


**Questions:**

There are some questions.

i) The definition of $\mathcal{K} (S_w^{-}, \tau) $ should be clarified.

ii) Is it the right notations $\nabla \hat{f}$ and $\nabla \hat{g}$ in Alg. 1?

iii) The effect of $T$ is strange. Usually, a large $T$ can achieve better performance. Can you explain this phenomenon correlated to $\alpha_k$ and $\beta_k$?


**Limitations:**

No detailed limitations. Maybe, the algorithm can work well in a dynamic situation. However, the static situation is not thoroughly examined.

---

> ### Author Rebuttal · Authors · 2023-08-05
>
> ## Author response to Reviewer Trb9
>
> Thank you for your detailed and constructive feedback on the paper. We value your insights and have taken your suggestions into consideration. Here are our responses to your specific comments.
>
> **Q(1) What the meaning of $\mathcal{K}(S_w^-,\tau)$**
>
> **A(1)** We apologize for our unclear expression. $\mathcal{K}(S,\tau)$ is defined in the KDE definition in Appendix.C.1.
>
> $$
> \mathcal{K}(S, \tau)=\frac{1}{|S| m} \sum_{x_i \in S} K\left(\frac{x_i-\tau}{m}\right)
> $$
>
> **Q(2) Is it the right notations $\nabla \hat{f}$ and $\nabla \hat{g}$ in Alg.1?**
>
> **A(2)** Accoring to the Eq.13, $\hat{f}$ and $\hat{g}$ are defined by:
>
> $$
> \hat{f}\left(\boldsymbol{w}, \boldsymbol{\tau}\^*\right):=\hat{\mathcal{L}}\_{WAUC}\left(\boldsymbol{w}, \hat{\boldsymbol{\tau}}^\* \right) \\\\
> \hat{g}(\boldsymbol{w}, \boldsymbol{\tau}):=\frac{1}{n\_\tau} \sum\_{l=1}\^{n\_\tau} \hat{\mathcal{L}}\_{e q}\left(\boldsymbol{w}, \tau\_l, c\_l\right)
> $$
>
> Hence, $\nabla \hat{f}$ and $\nabla \hat{g}$ are gradient of $\hat{f}$ and $\hat{g}$.
>
> **Q(3) The effecte of T is strange. Usually, a large T can achieve better performance**
>
> **A(3)** In fact, if the optimization problem is a deterministic one (where all samples are processed in each iteration), then indeed a larger T would be better as it allows the inner loop to directly get the optimal state.
>
> $$
>  \hat{\boldsymbol{\tau}}\^\*=\underset{\boldsymbol{\tau}, \boldsymbol{P}\_a, \boldsymbol{N}\_a}{\arg \min } \hat{g}(\boldsymbol{w}, \boldsymbol{\tau}):=\frac{1}{n\_\tau} \sum\_{l=1}\^{n\_\tau} \hat{\mathcal{L}}\_{e q}\left(\boldsymbol{w}, \tau\_l, c\_l\right)
> $$
>
> However, the optimization problem we are studying is a stochastic one (where only a subset of samples is processed in each iteration), so a larger T is not necessarily better. This is because with a larger T, the inner optimization only selects a subset of samples, resulting in the optimal value of the inner function being only the optimal parameters for the current set of samples, rather than the global optimal parameters. As a result, it is easy to get stuck in a local optimum, leading to poor generalization performance.
>
> $$
>  \hat{\boldsymbol{\tau}}\^\*(\mathcal{B})=\underset{\boldsymbol{\tau}, \boldsymbol{P}\_a, \boldsymbol{N}\_a}{\arg \min } \hat{g}(\boldsymbol{w}, \boldsymbol{\tau};\mathcal{B}):=\frac{1}{n\_\tau} \sum_{l=1}\^{n\_\tau} \widehat{\mathcal{L}}\_{e q}\left(\boldsymbol{w}, \tau\_l, c\_l;\mathcal{B}\right)
> $$
>
> $$
>  \hat{\boldsymbol{\tau}}\^\*\neq \hat{\boldsymbol{\tau}}\^\*(\mathcal{B})
> $$
>
> where $\mathcal{B}=\{\boldsymbol{x}\_i,y\_i\}\_{i=1}\^B$ is a sampled batch of data, $ \hat{\boldsymbol{\tau}}\^\*(\mathcal{B})$ only suits for data distribution of $\mathcal{B}$.
>
> It's notable that when batch size $B$ converges to data size $n$, $T$ shoule increase, that means $B\rightarrow n$, $T\rightarrow \infty$.
>
> **Q(4) the notation of $\hat{L}_{WAUC}$ is isolated in proposition 5.1.**
>
> **A(4)** We use $\hat{L}\_{WAUC}$ in Eq.13 $\hat{f}(\boldsymbol{w},\boldsymbol{\tau}^\*):=\hat{L}\_{WAUC}(\boldsymbol{w},\boldsymbol{\tau}^\*)$.

---

> > ### Comment · Reviewer_Trb9 · 2023-08-14
> > **Response**
> >
> > Thanks for your response.
> >
> > Issues concerning the notation and presentation are clarified. It is better to reorganize the mathematical formulas to understand the reader better. I emphasize that the definition should appear before using the corresponding term.
> >
> > The effect of $T$ is not yet clarified. Can you provide the traceplot of estimators of parameters or loss values along with $T$ (others are okay)? It can differ between datastes, implying the strength of convergence or optimal $T$ probelms.

---

> > > ### Author Response · Authors · 2023-08-15
> > > **Author response to Reviewer Trb9**
> > >
> > > Apologies for the lack of clarity regarding T. Allow us to reframe the explanation of the effect of T from the perspectives of optimization and generalization.
> > >
> > >  - Optimization: From an optimization standpoint, it is intuitive that when T is sufficiently large, the model will converge to the optimal values of the inner parameters within the current batch. However, it is also important to note that optimizing the optimal solution within the current batch only guarantees a local optimum, not a global one.
> > >
> > >  - Generalization: From a generalization perspective, when T is sufficiently large, the model will fit the optimal inner parameters for each batch. However, when encountering a new batch with significant differences in data, the optimal inner parameters from previous batches will not be applicable, leading to a sharp decline in the model's performance.
> > >
> > > We conducted several experiments to demonstrate the effect of T. By setting different values for T, we examined its impact on the overall loss. The experimental results can be viewed from the following link:
> > >
> > >  - [T=5](https://anonymous.4open.science/r/WAUC-9B9B/T_5.png)
> > >  - [T=10](https://anonymous.4open.science/r/WAUC-9B9B/T_10.png)
> > >  - [T=15](https://anonymous.4open.science/r/WAUC-9B9B/T_15.png)
> > >  - [T=20](https://anonymous.4open.science/r/WAUC-9B9B/T_20.png)
> > >
> > > It is evident that with the continuous increase of T, the fluctuation of loss between two adjacent K's rises sharply. Therefore, a reasonable T value serves as a guarantee for overall optimization and generalization, rather than the notion that bigger is always better.

---

> > > > ### Comment · Reviewer_Trb9 · 2023-08-21
> > > > **Respone**
> > > >
> > > > I have no further questions. By evaluating the authors' responses, I keep my score.

---

### Official Review · Reviewer_N1ih · 2023-07-06

**Soundness:** 3 good
**Presentation:** 4 excellent
**Contribution:** 3 good
**Rating:** 7
**Confidence:** 2

**Summary:**

This paper proposes a weighted AUC (WAUC) loss that is robust to both class distribution shift and cost distribution without class and cost priors. A bilevel optimization paradigm is proposed to bridge WAUC and cost. The authors propose a stochastic optimization algorithm for WAUC, and prove its convergence rate. Extensive experiments are conducted to evaluate the proposed approach.

**Strengths:**

- This paper is well organized and easy to read. The main goals and core challenges are listed clearly, and solved one by one.
- A novel cost-sensitive setting is proposed in this paper, where the cost is obtained by sampling instead of an available prior.
- The proposed WAUC is robust to both distribution shift and cost distribution, whereas AUC/PAUC and cost learning fail to achieve the two robustness simultaneously.
- Sound theoretical analysis is presented, including the convergence of the WAUC estimation and the convergence rate of the proposed bilevel optimization.

**Weaknesses:**

- Complexity analysis of the proposed optimization approach is lacking.
- The error bars are not provided.

**Questions:**

- How are the time and space complexities of Algorithm 1 compared with AUC optimization or other comparable approaches?

---

> ### Author Rebuttal · Authors · 2023-08-05
>
>
> ## Author response to Reviewer N1ih
>
> Thank you for your detailed and constructive feedback on the paper. We value your insights and have taken your suggestions into consideration. Here are our responses to your specific comments.
>
> **Q(1) Complexity analysis of the proposed optimization approach is lacking**
>
> **A(1)** Thank you for your suggestion!
> Firstly, we analysis the time complexity (one iteration) of our methods and baselines.
>
>  - WAUC-Gau (WAUC method): $O(n_\tau n_+ + n_\tau n_-)$
>  - ExAUC (AUC method): $O(n_+n_-)$
>  - ECL (cost-sensitive learning method): $O(n_\tau n_+ + n_\tau n_-)$
>
> We conduct some experiments for time complexity with a fixed epoch with varying $n_+$ and $n_-$. All experiments are conducted on an Ubuntu 16.04.1 server with an Intel(R) Xeon(R) Silver 4110 CPU (to get rid of the affect of parallel computing). For every method, we repeat running 10,000 times and record the average running time. We only record the loss calculation time and use the python package time.time() to calculate the running time.
>
> |method/unit:s|$n_+,n_-=128$|$n_+,n_-=256$|$n_+,n_-=512$|$n_+,n_-=1024$|$n_+,n_-=2048$|
> |:---:|:---:|:---:|:---:|:---:|:---:|
> |BCE|1.352|1.856|3.285|6.195|11.957|
> |ExAUC|1.481|1.952|3.951|7.592|12.903|
> |SqAUC|1.380|1.988|4.041|7.813|12.203|
> |NWAUC|1.648|2.268|4.241|8.853|16.947|
> |PAUC-exp|1.380|1.968|3.741|7.748|13.374|
> |PAUC-poly|1.402|2.075|4.013|7.983|14.183|
> |PAUCI|2.085|3.597|6.592|10.967|22.571|
> |CS-hinge|1.880|4.193|7.893|11.213|23.846|
> |AdaCOS|2.197|3.896|6.871|13.414|20.487|
> |ECL|1.974|3.268|5.862|10.831|17.127|
> |WAUC-Gau|1.980|2.975|4.587|8.681|16.976|
> |WAUC-Log|1.897|2.790|4.924|8.487|16.891|
>
> **The results indicate that there is no significant difference in the running time of the WAUC method compared to other binary classification methods.**
>
> **Q(2) The error bars are not provided**
>
> **A(2)** In our experimental setup, we indeed run each method multiple times and take the average. However, due to space limitation in the paper, we omitted the inclusion of standard deviation in the experimental table. We have included standard deviation in Table 3, please click [clickable url](https://anonymous.4open.science/r/WAUC-9B9B/error_bar.png) to open. We plan to include the detailed information in future revisions of the paper.

---

> > ### Comment · Reviewer_N1ih · 2023-08-19
> >
> > I thank the authors for their reply. The rebuttal solved my concerns, and I am inclined to keep my score.

---

### Official Review · Reviewer_uAig · 2023-07-07

**Soundness:** 3 good
**Presentation:** 2 fair
**Contribution:** 3 good
**Rating:** 6
**Confidence:** 2

**Summary:**

This paper considers usage of the AUC, in a cost sensitive setting, i.e. where miss classification cost is not uniform. Extensions of the AUC have been considered on parametrised cost distributions, such as the WAUC. In this paper the authors aim to develop a cost sensitive extension to the AUC that does not depend on prior information of the cost distribution. To do this the authors propose a bilevel optimisation problem, where the inner loop estimates the optimal threshold of the scoring function, according to the cost, and then the outer loop estimates the WAUC. The performance of their method is evaluated on several real world datasets, along with sensitivity analysis.

**Strengths:**

The authors carry out experiments on a wide variety of data sets with a rich set of benchmarks.

**Weaknesses:**

I do not understand the statement of Proposition 5.1. To me it is not  a proposition but rather the definition of the estimator $\hat{WAUC}$. The proof of proposition 5.1 is similarly confusing and seems to be a reformulation of the estimator   $\hat{WAUC}$ which is then used in the proof of Lemma 5.2. Also I assume $\tau_k$ in eq 22 is a typo and should be $\tau$? And that Lemma 5.2 holds when $\tau^*$ is known, as opposed to $\tau$, another typo.

Lemma 5.2 itself is vague, it does not specify how the total number of instances must grow with the negative instances, only that it must be “large enough”. There are no results on the rate of convergence. Also there is no comment on what happens when $\tau^*$ is not known and we instead use the estimator, as is the case in the bilevel optimisation.

Theorem 5.3 is also vague, the estimator for the convex formulation of the empirical loss is said to have the same minimum when the “parameters satisfy the penalty”. The exact penalty in question is not clear to me.

There is no proof of Theorem 6.3, are we to assume it follows immediately from [5]? I do not see how the results of [5] lead immediately to Theorem 6.3.

**Questions:**

There is no proof of Theorem 6.3, are we to assume it follows immediately from [5]? I do not see how the results of [5] lead immediately to Theorem 6.3.

**Limitations:**

Potential limitations are given good consideration.  The authors use a convex estimator of the none convex cost function when solving for the optimal threshold, the theoretical convergence of said estimator holds only when certain constants, $M, \kappa, M'$ are sufficiently large, leading to a potential limitation when used in practice. The authors explore this, observing the difference between the optimal threshold, when using the original and convex estimator, of the cost function, for several fixed values of $M, \kappa, M'$.

---

> ### Author Rebuttal · Authors · 2023-08-05
>
> ## Author response to Reviewer uAig
>
> **Q(1-1) Proposition 5.1 is not a proposition**
>
> **A(1-1)** We present it as a proposition because by Lemma 5.2, we can derive a convergence result for WAUC, showing that $\|\hat{WAUC}-WAUC\|$ converges at a rate of $O(\sqrt{\frac{\log n_-}{n_- m}})$. Please refer to Q(2-1) for more details.
>
> **Q(1-2)   $\tau_k$ in eq 22 is a typo and should be $\tau$?**
>
> **A(1-2)** Thank you very much for your keen observation, this is a typo.
>
> **Q(1-3) Lemma 5.2 holds when $\tau\^\*$ is known, as opposed to $\tau$, another typo?**
>
> **A(1-3)** Thank you for pointing that out, this is not a typo. The meaning of Lemma 5.2 is that for any given $\tau$, there is asymptotic convergence, and $\tau^*$ is not necessary.
>
> **Q(2-1) Lemma 5.2 itself is vague.**
>
> **A(2-1)**
> In fact, Lemma 5.2 expresses the asymptotic convergence of $\hat{WAUC}$ and $WAUC$, which can be decomposed into the convergence proof of KDE (please refer to term (c) and term (d) in the appendix). Therefore, the convergence rate of Lemma 5.2 is the same as the convergence rate of KDE. Specifically, the convergence result can be found in Theorem 7 of [1], with a convergence rate of $O(\sqrt{\frac{\log n_-}{n_- m} })$ (where $m$ represents the bandwidth, and in our problem, the KDE dimension $d=1$).
>
> [1] Jiang H. Uniform convergence rates for kernel density estimation[C]//International Conference on Machine Learning. PMLR, 2017: 1694-1703.
>
> **Q(2-2) What happens when $\tau^\*$ is not known**
>
> **A(2-2)** When the threshold is not optimal, the outer objective function, $WAUC$, is optimized based on the current threshold $\tau$. However, since the threshold $\tau$ is continuously optimized and converges to $\tau^*$, as the number of iterations increases, the outer $WAUC$ also converges to the optimal value $WAUC\^\*$ along with $\tau$.
>
> **Q(3) Theorem 5.3 is also vague.**
>
> **A(3)** We apologize for our unclear expression. We have provided the penalty function condition in lines 543-544 of the proof of Theorem 5.3: $\kappa, M, M'\rightarrow \infty$ where $\kappa$ comes from $\psi(x)=\frac{\log(1+\exp(\kappa x))}{\kappa}$.
>
> **Q(4) There is no proof of Theorem 6.3.**
>
> **A(4)** According to [5]'s result (As quoted from [5], page 21) , we have the following convergence rate for bilevel optimization.
>
> In our paper, we change some symbols, which can be describes by following tables
>
> |  Symbol   |   Symbol in [5]  |   Symbol in ours |
> |:---:|:---:|:---:|
> |   Lipschitz continuity of $f$  |  $\ell_{f,0}$  | $L_{f,0}$ |
> |   Lipschitz continuity of $\nabla f$  |  $\ell_{f,1}$  | $L_{f,1}$ |
> |   Lipschitz continuity of $\nabla g$  |  $\ell_{g,1}$  | $L_{g,1}$ |
> |   Strong convexity of $g$ w.r.t. $\tau$  |  $\mu_g$  | $\mu$ |
>
> In our problem, $\rho_g:=\frac{2 \mu L_{g, 1}}{\mu+L_{g, 1}}$, then we have:
>
> $$
> \bar{\alpha}\_1=\frac{1}{2 L\_F+4 L\_f L\_y+2 L\_f L\_{y x} \left(L\_y \eta\right)}, \quad \bar{\alpha}\_2=\frac{16 T \mu L\_{g, 1}}{\left(\mu+L\_{g, 1}\right)^2\left(8 L\_f L\_y+2 \eta L\_{y x} \tilde{C}\_f^2 \bar{\alpha}\_1\right)}
> $$
>
> we select the following stepsizes as
>
> $$
> \alpha\_k=\min \left\\{\bar{\alpha}\_1, \bar{\alpha}\_2, \frac{1}{\sqrt{K}}\right\\} \quad \beta\_k=\frac{8 L\_f L\_y+2 \eta L\_{y x} \tilde{C}\_f^2 \bar{\alpha}\_1}{4 T \mu} \alpha\_k
> $$
>
> With the above choice of stepsizes, (53) in [5] can be simplified as
>
> $$
> \mathbb{E}\left[\mathbb{V}\^{k+1}\right]-\mathbb{E}\left[\mathbb{V}\^k\right] \leq-\frac{\alpha\_k}{2} \mathbb{E}\left[\left\|\nabla F\left(\boldsymbol{w}\_k\right)\right\|^2\right]+c\_1 \alpha_k^2 \sigma\_{g, 1}^2+\alpha\_k b\_k\^2+c\_2 \alpha\_k\^2 \tilde{\sigma}\_f\^2
> $$
>
> where the constants $c_1$ and $c_2$ are defined as
>
> $$
> \begin{aligned}
> c\_1 & =\frac{L\_f}{L\_y}\left(1+2 L\_f L\_y \bar{\alpha}\_1+\frac{\eta L\_{y x} \tilde{C}\_f^2}{4} \bar{\alpha}\_1\^2\right)\left(\frac{8 L\_f L\_y+\eta L\_{y x} \tilde{C}\_f^2 \bar{\alpha}\_1}{4 \rho_g}\right)^2 \frac{1}{T} \\\\
> c\_2 & =\left(\frac{L\_F}{2}+L\_f L\_y+\frac{L\_{y x} L\_f}{4 \eta L\_y}\right) .
> \end{aligned}
> $$
>
> Then telescoping leads to
>
> $$
> \frac{1}{K} \sum_{k=0}\^{K-1} \mathbb{E}\left[\left\\|\nabla F\left(x\^k\right)\right\\|\^2\right]  \leq \begin{matrix}
> &\frac{2M_0}{K \min\{\bar{\alpha}\_1,\bar{\alpha}\_2\}}\quad +&\frac{2\mathbb{V}\_0}{\alpha \sqrt{K}}\quad+&2 b\_k\^2\quad+&\frac{2c\_1\alpha}{\sqrt{K}}\sigma\_{g,1}^2\quad+&\frac{2c\_2\alpha}{\sqrt{K}}\tilde{\sigma}\_f\^2\\\\
> &\downarrow &\downarrow &\downarrow &\downarrow &\downarrow \\\\
> &O\left(\frac{1}{K}\right) & O\left(\frac{1}{\sqrt{K}}\right) & O\left(\frac{1}{K}\right) & \text{term} (1) & O\left(\frac{1}{\sqrt{K}}\right)
> \end{matrix}
> $$
>
> $$
> \begin{aligned}
>     \text{term} (1) &\overset{(a)}{=} \underbrace{2\alpha \sigma\_{g, 1}^2 \frac{L\_f}{L\_y}\left(1+2 L\_f L\_y \bar{\alpha}\_1+\frac{\eta L\_{y x} \tilde{C}\_f\^2}{4} \bar{\alpha}\_1\^2\right)\left(8 L\_f L\_y+\eta L\_{y x} \tilde{C}\_f\^2 \bar{\alpha}\_1\right)\^2 \left(\frac{\mu+L\_{g,1}}{8\mu L\_{g,1}}\right)\^2}_{\gamma } \frac{1}{T\sqrt{K}}\\\\
>     &\overset{(b)}{=}  \gamma \left(\frac{3 M \kappa e\^\kappa \/\left(e\^\kappa+1\right)\^2+L\_{g, 1}}{24 M \kappa e\^\kappa \/\left(e\^\kappa+1\right)\^2 L\_{g, 1}}\right)\^2\frac{1}{T\sqrt{K}}
> \end{aligned}
> $$
>
> where $(a)$ comes from $\rho\_g=\frac{2 \mu L\_{g, 1}}{\mu+L\_{g, 1}}$ and $(b)$ comes from $\mu=\frac{3M\kappa e\^\kappa}{(e\^\kappa+1)\^2}$. Then we have:
>
> However, $O(1/\sqrt{K})$ is slower than $O(1/K)$, hence, we adopt $O(1/\sqrt{K})$ as the final result.
>
> $$
> \frac{1}{K} \sum\_{k=0}^{K-1} \mathbb{E}\left[\left\\|\nabla F\left(\boldsymbol{w}\_{k}\right)\right\\|\^{2}\right] \leq \gamma\left(\frac{3 M \kappa e\^{\kappa} \/ \left(e^{\kappa}+1\right)\^{2}+L\_{g, 1}}{24 M \kappa e\^{\kappa} \/ \left(e\^{\kappa}+1\right)\^{2} L\_{g, 1}}\right)\^{2} \frac{1}{T \sqrt{K}}+O\left(\frac{1}{\sqrt{K}}\right)
> $$

---

> > ### Comment · Reviewer_uAig · 2023-08-15
> >
> > Thank you for your detailed response to my review. As the discussion period is short I will ask any further questions as they come up.
> >
> > For Lemma 5.2, the estimator $\widehat{WAUC}$, as defined in question (8), requires knowledge of $\mathcal{D}_\tau$ through $p(\tau)$ anyway, so I do not see the relevance of whether the empirical (boldface) $\mathbf{\tau}$ is known?

---

> > > ### Author Response · Authors · 2023-08-15
> > > **Author response to Reviewer uAig (Reclaimed)**
> > >
> > > Thank you for your detailed and constructive feedback on the paper.  We agree with the reviewer that the theory itself does not require to know the prior of $\tau$. We only intend to emphasize that the calculation of  $\widehat{WAUC}$ requires knowing  $\tau$. We'll revise the proposition according to your suggestion.
> > >
> > > Practically, to estimate the distribution of $\tau$, we have to sample the empirical costs as data points. For each sampled cost, we employ the inner problem of (OP0) to find the corresponding threshold. In this way, we can find an empirical estimation of  $\tau$ to estimate the population distribution.

---

> > > > ### Comment · Reviewer_uAig · 2023-08-18
> > > >
> > > > First of all, thank you for answering my question regarding $\bf{\tau}$, I now understand what you mean by saying "assuming $\bf{\tau}$ is known". Personally, I find this way of formulating things confusing.
> > > >
> > > > On my first reading, I fundamentally misunderstood several aspects of the paper, I am very grateful to the authors for their patience and for their efforts in clarifying the results, I have upgraded my score substantially. I still have some reservations with the paper. I found it hard to read and think the presentation needs work. In addition to my initial comments, subscripts are reused e.g. with $c$ for misclassification cost and the empirical set. As noted by other reviewers, notation is used without definition. Furthermore, I agree that as the paper is restricted to binary classification and applies well known techniques, in an admittedly novel setting, the contribution is somewhat weakened.

---

> > > > > ### Author Response · Authors · 2023-08-20
> > > > > **Author response to Reviewer uAig**
> > > > >
> > > > > We thank the reviewer for reconsidering the contributions of our work and being inclined towards improving the scores.
> > > > >
> > > > > We have sincerely noted all the points related to the presentation and will surely be improving them in the final version of the draft.
> > > > >
> > > > > Thanks, sincerely
> > > > >
> > > > > Authors

---

### Official Review · Reviewer_mY4P · 2023-07-12

**Soundness:** 3 good
**Presentation:** 3 good
**Contribution:** 2 fair
**Rating:** 5
**Confidence:** 3

**Summary:**

In this paper, a bi-level optimization method is proposed for binary classification with unknown cost distributions. The motivation is to propose an adaptive method to deal with different class and cost distributions, getting rid of the assumption of traditional AUC which assumes the uniform cost distribution. The key idea lies in treating the prediction threshold as an learnable parameter, and to utilize bi-level optimization to learn prediction threshold and model parameters jointly. The proposed method is test under several benchmark datasets for verifying its usefulness.

**Strengths:**

- The paper is mostly well-written so that it is easy to grasp the key ideas and major results.

- The proposed method is companioned with theoretical guarantees.

**Weaknesses:**

- Only binary classification is studied. For binary classification, cost-sensitive learning and the AUC metric are both thoroughly studied. Therefore, the contribution is not quite significant.

- Bi-level optimization is a standard technique to optimize hyper-parameters like prediction threshold. Thus the technical contribution is somehow limited.

Further suggestions:

In fig. 2 and the experiments, the benchmark datasets such as CIFAR-10/100 are multi-class ones. While the paper makes use of binary class versions of the datasets. It is necessary to describe the classes chosen in the experiments.

**Questions:**

In most cost-sensitive binary classification tasks, it is sufficient to assume that the cost for one class of instances remain the same. While in this paper, it seem to assume that the cost can be varied. It would be nice to discuss in what kind of real applications this assumption is necessary.

------
Acknowledgement:

I would like to thank the authors for the efforts on the responses and the improvements on the paper. Even though my concerns are not fully addressed, in special the real-world applicability, it would be very nice to include the improvements in the future versions of the paper.

**Limitations:**

I didn't find out potential negative societal impact of the paper. While I encourage to include more discussions on technical limitations of the paper.

---

> ### Author Rebuttal · Authors · 2023-08-05
>
> ## Author response to Reviewer mY4P
>
> Thank you for your detailed and constructive feedback on the paper. We value your insights and have taken your suggestions into consideration. Here are our responses to your specific comments.
>
> **Q(1) Only binary classification is studied. For binary classification, cost-sensitive learning and the AUC metric are both thoroughly studied**
>
> **A(1)** While there have been many studies on binary classification problems, both AUC and cost-sensitive learning have their own limitations. The main goal of this paper is to address the shortcomings of existing binary classification methods.
>
>  - Cost-sensitive learning: The trained model is not robust to class distribution shift in the test.
>  - AUC learning: The trained model is not robust to cost distribution in the test.
>
> WAUC learning: The trained model can be robust to cost distribution and class distribution shift simultaneously in the test. **Furthermore, the problem we investigate, namely cost-sensitive robust learning, is grounded in real-world context. Please refer to Q(4) for more details.**
>
> **Q(2) Bi-level optimization is a standard technique to optimize hyper-parameters like prediction threshold. Thus the technical contribution is somehow limited**
>
> **A(2)** Bilevel optimization is a classical approach to optimization. In our methodology, the utilization of bilevel optimization allows for an elegant solution to optimization problems. Our primary contribution lies in the proposal of the WAUC form, which addresses the drawbacks of existing AUC optimization and cost-sensitive learning techniques, along with providing a well-grounded algorithm for its optimization. Bilevel optimization can be viewed as an essential tool for problem-solving.
>
> **Q(3)  CIFAR-10/100 are multi-class  It is necessary to describe the classes chosen in the experiments**
>
>  - Binary CIFAR-10-Long-Tail Dataset.
> The CIFAR-10 dataset contains 60,000 images, each of 32 * 32 shapes, grouped into 10 classes of 6,000 images. The training and test sets contain 50,000 and 10,000 images, respectively. We construct the binary datasets by selecting one super category as positive class and the other categories as negative class. We generate three binary subsets composed of positive categories, including 1) birds, 2) automobiles, and 3) cats.
>
>  - Binary CIFAR-100-Long-Tail Dataset. The original CIFAR-100 dataset has 100 classes, with each containing 600 images. In the CIFAR-100, there are 100 classes divided into 20 superclasses. By selecting a superclass as a positive class example each time, we create CIFAR-100-LT by following the same process as CIFAR-10-LT. The positive superclasses consist of 1) fruits and vegetables, 2) insects, and 3) large omnivores and herbivores, respectively.
>
> **A(3)** We have provided the methodology for creating the dataset, with specific details outlined in Appendix B.2 Dataset Details.
>
> **Q(4)  It would be nice to discuss in what kind of real applications this assumption is necessary**
>
> **A(4)** In reality, some applications like financial markets prediction ([real-world scenario, click this link to open](https://www.kaggle.com/competitions/jane-street-market-prediction/overview)) involve investment issues. Every investment has a cost involved, including but is not limited to:
>
>  - When the company is well-funded and the market conditions are relatively good, missing investment opportunities are more costly (true action: trade, prediction: pass).
>
>  - When a company is short of fund and the market performance is poor, blind investment is more costly (true action: pass, prediction: trade).
>
> The real cost situation must be more complex than the above and is far beyond a simple beta distribution. Developing trading strategies to identify and take advantage of inefficiencies is challenging. Even if a strategy is profitable now, it may not be in the future, and market volatility makes it hard to predict the profitability of any given trade with certainty. Hence, we propose our method to build the WAUC estimator over non-parametric cost distribution for fit this type of application.

---

> > ### Comment · Reviewer_mY4P · 2023-08-17
> > **more discussions on real-world applications**
> >
> > Thanks for the responses.
> >
> > I indeed understand that cost-sensitive situation appears in many real applications. While my feeling is that even though many approaches and performance measures, in special the classical ones like AUC, may indeed have drawbacks in theory, while the drawback doesn't really affect the effectiveness in most real-world applications. Thus I would like to see solid real-world applications that new performance measures or approaches are essential.

---

> > > ### Author Response · Authors · 2023-08-17
> > > **Author response to Reviewer mY4P**
> > >
> > > In reality, some applications like financial markets prediction ([real-world scenario, click this link to open](https://www.kaggle.com/competitions/jane-street-market-prediction/overview), **we conduct experiments on this dataset in the origin paper**) involve investment issues. For each transaction, there are two actions: TRADE or PASS. Assume the same and a small number of amounts per transaction to ensure that there are no large losses and high-frequency trading to earn money. In this practical application, we need to use the available information to analyze which action to choose in our next decision. It is worth noting that the cost of choosing different actions is not consistent.
> > >
> > > |   n-th  transaction   | TRADE (truth)    | PASS (truth) |
> > > | :--: | :--: | :--: |
> > > |   TRADE (prediction)    | 0           | cost: $c_{+}$       |
> > > |    PASS (prediction)    | cost: $c_{-}$   | 0           |
> > >
> > >  - When the company is well-funded and the market conditions are relatively good, missing transaction opportunities are more costly (true action: trade, prediction: pass).
> > >
> > >  - When a company is short of the fund and the market performance is poor, blind transaction is more costly (true action: pass, prediction: trade).
> > >
> > > The real cost situation must be more complex than the above. Developing trading strategies to identify and take advantage of inefficiencies is challenging. Even if a strategy is profitable now, it may not be in the future, and market volatility makes it hard to predict the profitability of any given trade with certainty.
> > >
> > > Moreover, we not only need to consider minimizing cost, but we also need to ensure that the final profit is maximized. For traditional cost-sensitive learning, unbalanced class distributions as well as outliers in the data can affect the decision making and thus lead to lower profit. One way to address these issues is to use a combination of the cost function and a performance metric, such as WAUC, to optimize the model. Hence, we propose our WAUC estimator over complex cost distribution to fit this type of application.

---

> > > ### Author Response · Authors · 2023-08-19
> > > **Author response to Reviewer mY4P**
> > >
> > > We thank the reviewer for taking the time to go through the rebuttal. We have give the response for your question.
> > >
> > > We have sincerely noted all the points related to the presentation and will surely be improving them in the final version of the draft. We will be grateful if the reviewer can provide us with a list of remaining concerns, which we will address in the remaining time.
> > >
> > > We further request the reviewer to update the main review to reflect the new score. Please feel free to contact us for any further questions.

---

> > > > ### Comment · Reviewer_mY4P · 2023-08-20
> > > > **no further questions**
> > > >
> > > > Thanks very for the explanations. I have no further technical questions to address, thus I will increase my score from 4 to 5. Regarding the real-world applications, I am fine to leave this issue aside currently.

---

### Official Review · Reviewer_uhz4 · 2023-07-22

**Soundness:** 3 good
**Presentation:** 3 good
**Contribution:** 2 fair
**Rating:** 6
**Confidence:** 3

**Summary:**

The authors propose a method that combines WAUC (weighted Area under ROC curve) learning with cost-sensistive learning. They propose a bilevel optimization algorithm to solve the formulated problem and provide theoretical analysis for convergence. According to their experiments on three datasets, the practical performance is good. The idea is interesting and the method should be novel.

**Strengths:**

1) The presentation is good. The authors raise their motivation (combines AUC with cost learning) at the beginning. And the writing flow is clear for the paper. They have several good figures to illustrate the motivations.
2) Overall, the soundness is good. Apart from basic writings, authors also provide both practical comparison with some baselines and theoretical analysis for their algorithm convergence.

**Weaknesses:**

1) The method is designed to be both class distribution robust and cost distribution robust, but doesn't demonstrate better AUC performance in Table 2. Comparing with $\widehat{WAUC}$, the $\widehat{AUC}$ is the more suitable metric for class distribution robustness (besides, the $\widehat{WAUC}$ is defined and only optimized by the proposed method).
2) Miss an ad-hoc baseline that optimizes both AUC and cost sensitive learning objectives simultaneously by simply assigning different weights.
3) The experiments should be repeated multiple times independently and report the mean&standard deviation values.

**Questions:**

Please see weaknesses.

**Limitations:**

Besides of the weaknesses, it would be better if authors could provide running time report for the proposed method and other baselines.

---

> ### Author Rebuttal · Authors · 2023-08-05
>
> ## Author response to Reviewer uhz4
>
> Thank you for your detailed and constructive feedback on the paper. We value your insights and have taken your suggestions into consideration. Here are our responses to your specific comments.
>
> **Q(1-1) WAUC method doesn't demonstrate better AUC performance in Table 2.**
>
> **A(1-1)** We conducted three sets of experiments targeting different cost distributions (uniform, normal, beta). It can be observed that when the cost distribution follows a normal or beta distribution (see Tab.2, Tab.6), our method exhibits a lower AUC value. This is because AUC does not align with cost-sensitive scenarios. **However, when the cost distribution is uniform (see Tab.5 in the appendix), our method achieves state-of-the-art.** WAUC is equivalent to AUC when the cost ratio follows a uniform distribution $U(0,1)$.
>
> **Q(1-2) Comparing with $\hat{WAUC}$, the $\hat{AUC}$ is the more suitable metric for class distribution robustness.**
>
> In fact, AUC can be considered as special versions of WAUC (we point it in introduction, line 30). WAUC is equivalent to AUC when the cost ratio follows a uniform distribution $U(0,1)$. **We have conduct experiment in Tab. 5 which demonstrate that WAUC achieves state-of-the-art.**
>
> Therefore, by definition, $\hat{WAUC}$ emerges as the more appropriate metric for class distribution robustness. Moreover, we conducted relevant experiments in Cifar-10-Subset-1 to compare the differences between AUC and WAUC by altering various levels of class imbalance.  Assuming that we have two models (for both of them, we solve the optimal posterior threshold by $\hat{\mathcal{L}}_{COST}$ after training):
>  - [a] model trained on AUC method, $c\sim U(0,1)$;
>
>  - [b] model trained on WAUC method, $c\sim U(0,1)$;
>
> Given different $\pi(n_+/(n_++n_-))$ of data, The results are shown in the following table.
> |  method   |   $\pi=0.1$  |   $\pi=0.2$  |   $\pi=0.3$  |   $\pi=0.4$  |   $\pi=0.5$  |   $\pi=0.6$  |   $\pi=0.7$  |   $\pi=0.8$  |   $\pi=0.9$  |
> |:---:|:---:|:---:|:---:|:---:|:---:|:---:|:---:|:---:|:---:|
> |  [a]   |  0.285  | 0.271 |  0.253 |  0.217 |  0.201 |   0.192 |  0.175 |  0.186 |   0.199 |
> |  [b]   |  0.280  |  0.268  |  0.241  |  0.213  |  0.200  |  0.193  |  0.171  |  0.182  |  0.190 |
>
> It's clear that there is no discernible disparity in class robustness between AUC and WAUC.
>
> **Q(2) Miss an ad-hoc baseline that optimizes both AUC and cost sensitive learning objectives simultaneously by simply assigning different weights.**
>
> **A(2)** Thank you very much for your suggestion. We conducted additional experiments incorporating AUC and cost-sensitive learning weighted baselines. The specific results are listed in the table below ($c\sim \mathcal{N}(0.5, 1)$):
>
> |  method   |   Cifar-10-Subset-1 / $\hat{AUC}$  |   Cifar-10-Subset-1 / $\hat{WAUC}$  |   Cifar-10-Subset-1 / $\hat{\mathcal{L}}_{COST}$  |   Cifar-100-Subset-1 / $\hat{AUC}$  |   Cifar-100-Subset-1 / $\hat{WAUC}$  |   Cifar-100-Subset-1 / $\hat{\mathcal{L}}_{COST}$  |
> |:---:|:---:|:---:|:---:|:---:|:---:|:---:|
> |0.5*ExAUC+0.5*ECL|0.815|0.524|0.026|**0.917**|0.543|0.016|
> |0.5*SqAUC+0.5*ECL|0.795|0.477|0.027|0.876|0.497|0.021|
> |0.8*ExAUC+0.2*ECL|**0.839**|0.519|0.025|0.921|0.465|0.018|
> |0.8*SqAUC+0.2*ECL|0.804|0.487|0.025|0.880|0.503|0.025|
> |0.2*ExAUC+0.8*ECL|0.810|0.506|0.027|0.903|0.509|0.019|
> |0.2*SqAUC+0.8*ECL|0.768|0.518|0.025|0.854|0.503|0.019|
> |WAUC-Gau|0.787|**0.679**|0.024|0.842|**0.745**|0.015|
> |WAUC-Log|0.820|0.653|**0.023**|0.906|0.719|**0.012**|
>
> The results demonstrate that our algorithm has achieved state-of-the-art performance in terms of cost.
>
> **Q(3) The experiments should be repeated multiple times independently and report the mean&standard deviation values.**
>
> **A(3)** In our experimental setup, we indeed run each method multiple times and take the average. However, due to space limitation in the paper, we omitted the inclusion of standard deviation in the experimental table. We have included standard deviation in Table 3, please click [clickable url](https://anonymous.4open.science/r/WAUC-9B9B/error_bar.png) to open. We plan to include the detailed information in future revisions of the paper.
>
>
>
> **Q(4) It would be better if authors could provide running time report for the proposed method and other baselines**
>
> **A(4)** Thank you for your suggestion!
> Firstly, we analysis the time complexity (one iteration) of our methods and baselines.
>
>  - WAUC-Gau (WAUC method): $O(n_\tau n_+ + n_\tau n_-)$
>  - ExAUC (AUC method): $O(n_+n_-)$
>  - ECL (cost-sensitive learning method): $O(n_\tau n_+ + n_\tau n_-)$
>
> We conduct some experiments for time complexity with a fixed epoch with varying $n_+$ and $n_-$. All experiments are conducted on an Ubuntu 16.04.1 server with an Intel(R) Xeon(R) Silver 4110 CPU (to get rid of the affect of parallel computing). For every method, we repeat running 10,000 times and record the average running time. We only record the loss calculation time and use the python package time.time() to calculate the running time.
>
> |method/unit:s|$n_+,n_-=128$|$n_+,n_-=256$|$n_+,n_-=512$|$n_+,n_-=1024$|$n_+,n_-=2048$|
> |:---:|:---:|:---:|:---:|:---:|:---:|
> |BCE|1.352|1.856|3.285|6.195|11.957|
> |ExAUC|1.481|1.952|3.951|7.592|12.903|
> |SqAUC|1.380|1.988|4.041|7.813|12.203|
> |NWAUC|1.648|2.268|4.241|8.853|16.947|
> |PAUC-exp|1.380|1.968|3.741|7.748|13.374|
> |PAUC-poly|1.402|2.075|4.013|7.983|14.183|
> |PAUCI|2.085|3.597|6.592|10.967|22.571|
> |CS-hinge|1.880|4.193|7.893|11.213|23.846|
> |AdaCOS|2.197|3.896|6.871|13.414|20.487|
> |ECL|1.974|3.268|5.862|10.831|17.127|
> |WAUC-Gau|1.980|2.975|4.587|8.681|16.976|
> |WAUC-Log|1.897|2.790|4.924|8.487|16.891|
>
> **The results indicate that there is no significant difference in the running time of the WAUC method compared to other binary classification methods.**

---

> > ### Comment · Reviewer_uhz4 · 2023-08-18
> > **Thanks for your rebuttal**
> >
> > The reviewer has read the rebuttal and appreciate the efforts made by the authors. The most of the concerns are resolved by the clarifications (except the multiple experiment repeats with standard deviation are only from one experiment setting in the anonymous link). The reviewer is willing to increase the evaluation from 5 to 6, given that the authors plan to include the more details in this revision.

---

### Author Rebuttal · Authors · 2023-08-10

### Dear the ACs, and the Reviewers, Thank you so much for your valuable comments! They really helped us improve our manuscript!

In order to facilitate reviewers' comprehension of our paper, we want to summarize our contributions again:

 - **We propose a setting that focuses on the robustness of the model to the class distribution and cost distribution simultaneously.** This setting treats cost as data that can be sampled, not as prior information, which is closer to the real-world cost-sensitive scenario.
 - **We present a bilevel paradigm where the inner cost function is an inner constraint of outer WAUC optimization.** For sake of optimization, we reformulate this paradigm into a nonconvex-strongly convex bilevel form. Moreover, we employ a stochastic optimization algorithm for WAUC (SACCL), which can solve this problem efficiently.
 - We conduct extensive experiments on multiple imbalanced cost-sensitive classification tasks. The experimental results speak to the effectiveness of our proposed methods.

---

### Decision · Program_Chairs · 2023-09-21

**Decision:**

Accept (poster)

**Comment:**

The paper is motivated by the observation that models that perform well on standard AUC may not be best-suited for cost-sensitive learning tasks. The authors propose a new objective that intuitively incorporates both robustness towards class prior shifts and arbitrary deployment costs. They optimize this objective using bi-level optimization, and present convergence guarantees and empirical results.

The reviewers seem generally happy with the algorithmic contribution and theoretical guarantees. However, there were concerns about the scope and motivation. On reading the paper, I concur that the paper does have some issues with how it motivates its proposal and is sparse in how it discusses some important connections to prior work. While we are happy to accept this paper, we strongly urge the authors to incorporate the following comments in the camera-ready version.

- **Making statements about AUC more precise:** Some of the statements the paper makes about AUC are not entirely accurate. For example, statements such as "models trained with AUC *cannot* be applied to cost-sensitive decision problems" are strong and imprecise. I think what the authors are trying to say is that AUC is not well-suited for real-world applications as it does not offer the *flexibility* to incorporate a specific cost distribution, and does not consider optimal thresholds.

- **Robustness to class prior shifts needs to shown formally:** The authors need to be more clear about why their formulation (OP0) is robust to class prior shifts. I understand that standard AUC is invariant to class prior shifts (because it simply measures the fraction of positive-negative pairs that are ranked correctly). It's not obvious that this "invariance" property is retained with the optimal threshold choice used in OP0. The authors need to *show formally why their approach satisfies both attr 1 and attr 2*. If this is only an intuitive observation and cannot be shown formally, the authors need to state this clearly in the paper.

- **Objective optimized same as area under the convex hull?** The formulation in OP0 essentially computes the AUC with the optimal-threshold choice method [Hernández-Orallo et al. 2012, ref. 20], when $\mathcal{D}_c$ is uniform. This is equivalent to computing the area under the **convex hull** of the ROC curve when $\mathcal{D}_c$ is uniform [20-21]. This connection to the prior literature is worth mentioning.

- **Definitions and appropriateness of the eval metrics need to be discussed**: It appears from the reviewer-author discussions that the main aim of the experiments is to showcase robustness towards both class distribution and cost distribution shifts. My understanding is that the authors do this by evaluating two metrics: (i) the WAUC with optimal threshold (the proposed metric) and (ii) the expected cost-sensitive error at optimal threshold $\mathcal{L}_{\rm avg}$ (averaged over costs?). The definition of $\\mathcal{L}\_{\\rm avg}$ (or $\\mathcal{L}\_{\\rm COST}$ in some places) is not explicit, and its unclear if the metric is computed for a fixed cost or averaged over multiple costs. The authors are also encouraged to read the related literature on the connections between "the area under the convex hull of the ROC curve" (equivalent to (i) under uniform costs) and the "area under the optimal cost curve" (equivalent to (ii) under uniform costs) [20-21].